

# Validation of the CrIS Fast Physical NH₃ Retrieval with ground-based FTIR

Enrico Dammers[1], Mark W. Shephard[2], Mathias Palm[3], Karen Cady-Pereira[4], Shannon Capps[5*], Erik Lutsch[6],
Kim Strong[6], James W. Hannigan[7], Ivan Ortega[7], Geoffrey C. Toon[8], Wolfgang Stremme[9], Michel Grutter[9],
Nicholas Jones[10], Dan Smale[11], Jacob Siemons[2], Kevin Hrpcek[12], Denis Tremblay[13], Martijn Schaap[14], Justus
Notholt[3], Jan Willem Erisman[1,15]

1. Cluster Earth and Climate, Department of Earth Sciences, Vrije Universiteit Amsterdam, Amsterdam, the
Netherlands
2. Environment and Climate Change Canada, Toronto, Ontario, Canada
3. Institut für Umweltphysik, University of Bremen, Bremen, Germany
4. Atmospheric and Environmental Research (AER), Lexington, Massachusetts, USA
5. Department of Mechanical Engineering, University of Colorado, Boulder, Colorado, USA
6. Department of Physics, University of Toronto, Toronto, Ontario, Canada
7. NCAR, Boulder, Colorado, United States
8. Jet Propulsion Laboratory, California Institute of Technology, Pasadena, California, USA
9. Centro de Ciencias de la Atmósfera, Universidad Nacional Autónoma de México, Mexico City, Mexico
10. University of Wollongong, Wollongong, Australia
11. National Institute of Water and Atmosphere, Lauder, New Zealand
12. University of Wisconsin-Madison Space Science and Engineering Center (SSEC), Madison, Wisconsin, USA.
13. Science Data Processing, Inc., Laurel, MD, United States
14. TNO Built Environment and Geosciences, Department of Air Quality and Climate, Utrecht, the Netherlands
15. Louis Bolk Institute, Driebergen, the Netherlands
*Now at Civil, Architectural, and Environmental Engineering Department, Drexel University, Philadelphia,
Pennsylvania, USA

*Correspondence to: E. Dammers (e.dammers@vu.nl)*

**Abstract**

Global reactive nitrogen emissions into the air have increased to unprecedented levels. Limiting the loss of

reactive nitrogen into the environment is one of the major challenges for humankind. At the current levels

ammonia (NH₃) is a threat to both the environment and human health. However, relatively little is known about

the total nitrogen budget and distribution around the world, due in part to the sparseness of observations over

most of the globe. Recent advances in the capabilities of measuring NH₃ with satellite instruments have

improved the situation with sensors such as the Infrared Atmospheric Sounding Interferometer (IASI) and the

Cross-Track Infrared Sounder (CrIS) making twice daily observations with global coverage. However, these

require validation to be truly useful, and one of the main challenges in the validation of the satellite NH₃ profile

and total column data products is the scarcity of measurements that can be directly compared. Presented here is

the validation of the CrIS Fast Physical Retrieval (CFPR) NH₃ column and profile measurements using ground-

based Fourier Transform Infrared (FTIR) observations. We use the total columns and profiles from seven FTIR

sites in the Network for the Detection of Atmospheric Composition Change (NDACC) to validate the satellite

data products. The overall FTIR and CrIS total columns compare well with a correlation of r = 0.77 (N=218)

with very little bias (a slope of 1.02). Binning the comparisons by total column amounts, for concentrations

larger than $1.0 \times 10^{16}$ molecules cm⁻², i.e. ranging from moderate to polluted conditions, the relative difference is

on average ~ 0 - 5% with a standard deviation of 25-50%, which is comparable to the estimated retrieval



uncertainties in both CrIS and the FTIR. For the smallest total column range where there are a large number of
observations at or near the CrIS noise level (detection limit) and the FTIR total columns are smaller than
$1.0 \times 10^{16}$ molecules $cm^{-2}$, the absolute differences between CrIS and the FTIR total columns are small with CrIS
showing a slight positive column bias around $+2.4 \times 10^{15}$ (standard deviation = $5.5 \times 10^{15}$) molecules $cm^{-2}$,
which corresponds to a relative difference of ~+50% (std = 100 %). The CrIS retrievals for these comparisons
typically show good vertical sensitivity down to ~850 hPa, and at this level the retrieved profiles also compare
well with the median absolute difference of 0.87 (±0.08) ppb and a corresponding median relative difference of
39 (±2)%. Most of the absolute and relative profile comparison differences are in the range of the estimated
retrieval uncertainties. However, the CrIS retrieval does tend to overestimate the concentrations in the levels
near the surface at low concentrations, most probably due to the detection limit of the instrument, and at higher
concentrations shows more of an underestimation of the concentrations in these lower levels.



## 1. Introduction

The disruption of the nitrogen cycle by the human creation of reactive nitrogen has created one of the major challenges for humankind (Rockström et al., 2009). Global reactive nitrogen emissions into the air have increased to unsurpassed levels (Fowler et al., 2013) and are currently estimated to be four times larger than pre-industrial levels (Holland et al., 1999). As a consequence the deposition of atmospheric reactive nitrogen has increased causing ecosystems and species loss (Rodhe et al 2002; Dentener et al., 2006; Bobbink et al., 2010). Ammonia ($NH_3$) as fertilizer is essential for agricultural production and is one of the most important reactive nitrogen species in the biosphere. $NH_3$ emission, atmospheric transport, and atmospheric deposition are major causes of eutrophication and acidification of soils and water in semi-natural environments (Erisman et al., 2008, 2011). Through reactions with sulphuric acid and nitric acid, ammonium nitrate and ammonium sulphate are formed which embody up to 50% of the mass of fine mode particulate matter ($PM_{2.5}$) (Seinfeld and Pandis., 1988; Schaap et al., 2004). $PM_{2.5}$ has been associated with various health impacts (Pope et al., 2002; 2009). At the same time, atmospheric aerosols impact global climate directly through their radiative forcing effect and indirectly through the formation of clouds (Adams et al., 2001; Myhre et al., 2013). By fertilizing ecosystems, deposition of $NH_3$ and other reactive nitrogen compounds also plays a key role in the sequestration of carbon dioxide (Oren et al., 2001).

Despite the significance and impact of $NH_3$ on the environment and climate, its global distribution and budget are still relatively uncertain (Erisman et al., 2007; Clarisse et al., 2009; Sutton et al., 2013). One of the reasons is that in-situ measuring of atmospheric $NH_3$ at ambient levels is complex due to the sticky nature and reactivity of the molecule, leading to large uncertainties and/or sampling artefacts with the currently used measuring techniques (von Bobrutzki et al., 2010; Puchalski et al., 2011). Measurements are also very sparse. Currently, observations of $NH_3$ are mostly available in north-western Europe and central North America, supplemented by a small number of observations made in China (Van Damme et al., 2015b). Furthermore, there is a lack of detailed information on its vertical distribution as only a few dedicated airborne measurements are available (Nowak et al., 2007, 2010; Leen et al., 2013, Whitburn et al., 2015, Shephard et al., 2015). The atmospheric lifetime of $NH_3$ is rather short, ranging from hours to a few days. In summary, global emission estimates have large uncertainties. Estimates of regional emissions attributed to source types different from the main regions are even more uncertain due to a lack of process knowledge and atmospheric levels (Reis et al., 2009).

Over the last decade the developments of satellite observations of $NH_3$ from instruments such as the Cross-track Infrared Sounder (CrIS, Shephard and Cady-Pereira, 2015), the Infrared Atmospheric Sounding Interferometer (IASI, Clarisse et al., 2009; Coheur et al., 2009; Van Damme et al., 2014a), the Atmospheric Infrared Sounder (AIRS, Warner et al., 2016), and the Tropospheric Emission Spectrometer (TES, Beer et al., 2008; Shephard et al., 2011) show potential to improve our understanding of the $NH_3$ distribution. Recent studies show the global distribution of $NH_3$ measured at a twice daily scale (Van Damme et al., 2014a, Van Damme et al., 2015a) and reveal seasonal cycles and distributions for regions where measurements were unavailable until now. Comparisons of these observations to surface observations and model simulations, show underestimations of the modelled $NH_3$ concentration levels, pointing to underestimated regional and national emissions (Clarisse et al.,



2009; Shephard et al., 2011; Heald et al., 2012; Nowak et al., 2012; Zhu et al., 2013; Van Damme et al., 2014b;
Lonsdale et al., 2016; Schiferl et al., 2014, 2016; Zondlo et al., 2016). However, the uncertainty of the satellite
observations is still high due to a lack of validation. The few validation studies showed a limited vertical, spatial
and or temporal coverage of surface observations to do a proper uncertainty analysis (Van Damme et al., 2015b;
Shephard et al., 2015; Sun et al., 2015). A recent study by Dammers et al. (2016a) explored the use of Fourier
transform infrared (FTIR-$NH_3$, Dammers et al., 2015) observations to evaluate the uncertainty of the IASI-$NH_3$
total column product. The study showed the good performance of the IASI-LUT (Look up table, LUT, Van
Damme et al., 2014a) retrieval with a high correlation (r ~ 0.8) but indicated an underestimation of around 30%
due to potential assumptions of the shape of the vertical profile (Whitburn et al., 2016, IASI-NN (Neural
Network, NN)), uncertainty in spectral line parameters and assumptions on the distributions of interfering
species. The study showed the potential of using FTIR observations to validate satellite observations of $NH_3$ but
also stressed the challenges of validating retrievals that do not provide the vertical measurement sensitivity, such
as the IASI-LUT retrieval. Since no IASI satellite averaging kernels are provided for each retrieval, and thus no
information is available on the vertical sensitivity and/or vertical distribution of each separate observation, it is
hard to determine the cause of the discrepancies between both observations.

The new CrIS Fast Physical Retrieval (Shephard and Cady-Pereira, 2015) uses an optimal estimation retrieval
approach that provides the information content and the vertical sensitivity (derived from the averaging kernels,
for more details see Shephard and Cady-Pereira, 2015), and robust and straightforward retrieval error estimates
based on retrieval input parameters. The quality of the retrieval has so far not been thoroughly examined against
other observations. Shephard and Cady-Pereira (2015) used Observing System Simulation Experiment (OSSE)
studies to evaluate the initial performance of the CrIS $NH_3$ retrieval, and report a small positive retrieval bias of
6% with a standard deviation of ±20% (ranging from ±12 to ±30% over the vertical profile). Note that no
potential systematic errors were included in these OSSE simulations.  Their study also shows good qualitative
comparisons with the Tropospheric Emission Spectrometer (TES) satellite (Shephard et al., 2011) and the
ground-level in situ Quantum Cascade-Laser (QCL) observations (Miller et al., 2014) for a case study over the
Central Valley in CA, USA, during the DISCOVER-AQ campaign.  However, currently there has not been an
extensive validation of the CrIS $NH_3$ retrievals using direct comparisons against vertical profile observations. In
this study we will provide both direct comparisons of the CrIS retrieved profiles against ground-based FTIR
observations, and comparisons of CrIS total column values against the FTIR and IASI.





## 2. Methods

### 2.1 The CrIS Fast Physical Retrieval

CrIS was launched in late October 2011 on board the Suomi NPP platform. CrIS follows a sun-synchronous orbit with a daytime overpass time at 13:30 local time (ascending) and a night time equator overpass at 1:30. The instrument scans along a 2200 km swath using a 3 x 3 array of circular shaped pixels with a diameter of 14 km at nadir for each pixel, becoming larger ovals away from nadir. In this study we use the $NH_3$ retrieval as described by Shephard and Cady-Pereira (2015). The retrieval is based on an optimal estimation approach (Rodgers, 2000) that minimizes the differences between CrIS spectral radiances and simulated forward model radiances computed from the Optimal Spectral Sampling (OSS) OSS-CrIS (Moncet et al., 2008), which is built from the well-validated Line-By-Line Radiative Transfer Model (LBLRTM) (Clough et al., 2005; Shephard et al., 2009; Alvarado et al., 2013). The fast computational speed of OSS facilitates the operational production of CrIS retrieved (Level 2) products using an optimal estimation retrieval approach (Moncet et al., 2005). The CrIS OSS radiative transfer forward model computes the spectrum for the full CrIS LW band, at the CrIS spectral resolution of $0.625$ $cm^{-1}$ (Tobin, 2012), thus the complete $NH_3$ spectral band (near 10 µm) is available for the retrievals. However, only a small number of micro windows are selected for the CrIS retrievals to both maximize the information content and minimize the influence of errors. Worden et al., (2004) provides an example of a robust spectral region selection process that takes into consideration both the estimated errors (i.e. instrument noise, spectroscopy errors, interfering species, etc.) and the associated information content in order to select the optimal spectral regions for the retrieval. The a-priori profiles selection for the optimal estimation retrievals follows the Tropospheric Emission Spectrometer (TES) retrieval algorithm (Shephard et al., 2011); Based on the relative $NH_3$ signal in the spectra the a-priori is selected from one of three possible profiles representing unpolluted, moderate, and polluted conditions. The initial guess profiles are also selected from these three potential profiles.

An advantage of using an optimal estimation retrieval approach is that averaging kernels (sensitivity to the true state) and the estimated errors of the retrieved parameter are computed in a robust and straight-forward manner (for more details see Shephard and Cady-Pereira, 2015). The total satellite retrieved parameter error is expressed as the sum of the smoothing error (due to unresolved fine structure in the profile), the measurement error (random instrument noise in the radiance spectrum propagated to the retrieval parameter), and systematic errors from uncertainties in the non-retrieved forward model parameters and cross-state errors propagated from retrieval-to-retrieval (i.e. interfering species) (Worden et al., 2004). As of yet we have not included error estimates for the systematic errors. The CrIS smoothing error is computed, but since in these FTIR comparison results we apply the FTIR observational operator (which accounts for the smoothing error), the smoothing error contribution is not included in the CrIS errors reported in the comparisons. Thus, only the measurement errors





are reported for observations used here; these errors can thus be considered the lower limit on the total estimated
CrIS retrieval error.
Figure 1 shows an example of CrIS NH$_3$ observations surrounding one of the ground-based FTIR instruments.
This is a composite map of all days in Bremen with observations in 2015. This figure shows the wide spread
elevated amounts of NH$_3$ across north-western Germany as observed by CrIS.

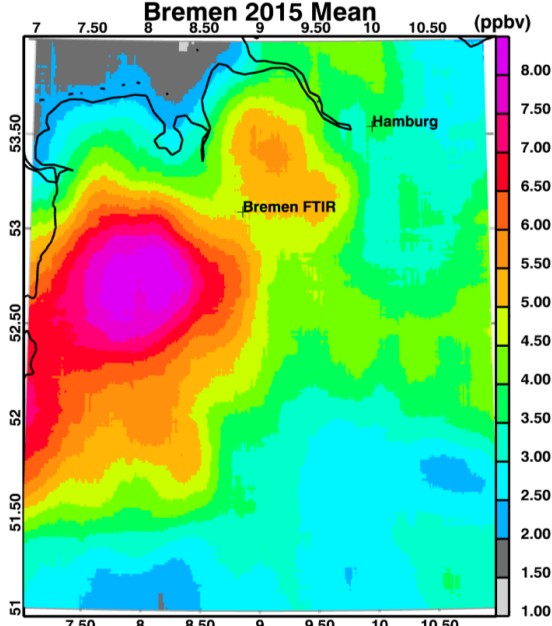

**Figure 1.** Annual mean of the CrIS retrieved NH$_3$ surface VMR values around the Bremen FTIR site for 2015.
Since the goal of this analysis is to evaluate the CrIS retrievals that provide information beyond the a-priori, we
only performed comparisons when the CrIS spectrum presents a NH$_3$ signal. We also focused our efforts on
FTIR stations that have FTIR observations with total columns larger than 5 x 10$^{15}$ molecules cm$^{-2}$ (~1-2 ppb
surface VMR). This restriction does mean that a number of sites of the FTIR-NH$_3$ dataset will not be used. For
comparability of this study to the results of the IASI-LUT evaluation in an earlier study by Dammers et al.,
(2016a) we include a short paragraph on the performance of the IASI-LUT and the more recent IASI-NN
product when applying similar constraints.
**2.2      FTIR-NH$_3$ retrieval**
The FTIR-NH$_3$ product used in this study is similar to the set described in Dammers et al. (2016a) and is based
on the retrieval methodology described by Dammers et al. (2015). The retrieval methodology uses two spectral
micro-windows whose spectral width depends on the NH$_3$ background concentration determined for the
observation stations and location (wider window for stations with background concentrations less than one ppb).
NH$_3$ is retrieved by fitting the spectral lines in the two micro-windows MW1 [930.32-931.32 cm$^{-1}$ or wide:




929.40-931.40 cm$^{-1}$] and MW2 [962.70-970.00 cm$^{-1}$ or wide: 962.10-970.00 cm$^{-1}$. An optimal estimation
approach (Rodgers et al., 2000) is used, implemented in the SFIT4 algorithm (Pougatchev et al., 1995; Hase et
al., 2004, 2006). There are a number of species that can interfere to some extent in both windows, with the
major species being $H_2O$, $CO_2$ and $O_3$ and the minor species $N_2O$, $HNO_3$, CFC-12, and $SF_6$. The HITRAN 2012
database (Rothman et al., 2014) is used for the spectral lines. A further set of spectroscopic line parameter
adjustments are added for $CO_2$ taken from the ATMOS database (Brown et al., 1996) as well as a set of pseudo-
lines for the broad absorptions by the CFC-12 and $SF_6$ molecules (created by NASA-JPL, G.C. Toon,
http://mark4sun.jpl.nasa.gov/pseudo.html). The $NH_3$ a-priori profiles are based on balloon measurements (Toon
et al., 1999) and refitted to match the local surface concentrations (depending on the station either measured or
estimated by model results). For the interfering species a-priori profiles we use the Whole Atmosphere
Community Climate Model (WACCM, Chang et al., 2008, v3548). The estimated errors in the FTIR-$NH_3$
retrievals are in the order of ~30% (Dammers et al., 2015) with the uncertainties in the $NH_3$ line spectroscopy
being the most important contributor. Based on the data requirements in section 2.1, a set of seven stations is
used (Table 1). For all sites except Wollongong in Australia we use the basic narrow spectral windows. For
Wollongong the wide spectral windows are used. For a more detailed description of each of the stations see the
publications listed in Table 1 or Dammers et al. (2016a).
**Table 1.** The location, longitudinal and latitudinal position, altitude above sea level, and type of instrument for
each of the FTIR sites used in this study. In addition, a reference is given to a detailed site description, when
available.

| Station | Lon (degrees) | Lat (degrees) | Altitude (m.a.s.l) | FTIR instrument | Reference |
|---|---|---|---|---|---|
| Bremen, Germany | 8.85E | 53.10N | 27 | Bruker 125 HR | Velazco et al., 2007 |
| Toronto, Canada | 79.60W | 43.66N | 174 | ABB Bomem DA8 | Wiacek et al., 2007 Lutsch et al., 2016 |
| Boulder, United States | 105.26W | 39.99N | 1634 | Bruker 120 HR | |
| Pasadena, United States | 118.17W | 34.20N | 350 | MkIV_JPL | |
| Mexico City, Mexico | 99.18W | 19.33N | 2260 | Bruker Vertex 80 | Bezanilla et al., 2014 |
| Wollongong, Australia | 150.88E | 34.41S | 30 | Bruker 125 HR | |
| Lauder, New Zealand | 169.68E | 45.04S | 370 | Bruker 120 HR | Morgenstern et al., 2012 |


### 2.3    IASI-NH₃

The CrIS retrieval will also be compared with corresponding IASI/FTIR retrievals using results from a previous
study by Dammers et al. (2016a). Both the IASI-LUT (Van Damme et al., 2014a) and the IASI-NN (Neural
Networks, Whitburn et al., 2016) retrievals from observations by the IASI instrument aboard MetOp-A will be
used. A short description of both IASI retrievals is provided here, for a more in-depth description, see the
respective publications by Van Damme et al. (2014a) and Whitburn et al. (2016). The IASI instrument on board
the MetOp-A platform is in a sun-synchronous orbit and has a daytime overpass at around 9:30 local solar time
and a night time overpass at around 21:30.  The instrument has a circular footprint of about 12 km diameter for
nadir viewing angles with of nadir observations along a swath of 2100 km. Both IASI retrievals are based on the
calculation of a dimensionless spectral index called the Hyperspectral Range Index (HRI) (Van Damme et al.,
2014a). The HRI is representative of the amount of $NH_3$ in the measured column. The IASI-LUT retrieval
makes a direct conversion of the HRI to a total column density with the use of a look-up-table (LUT). The LUT
is created using a large number of simulations for a wide range of atmospheric conditions which links the
Thermal Contrast (TC, the difference between the air temperature at 1.5 km altitude and the temperature of the
Earth surface) and the HRI to a $NH_3$ total column density. The retrieval includes a retrieval error based on the
uncertainties in the initial HRI and TC parameters. The more recent IASI-NN retrieval (Whitburn et al., 2016)
follows similar steps but it makes use of a neural network. The neural network combines the complete
temperature, humidity and pressure profiles for a better representation of the state of the atmosphere. At the
same time the retrieval error estimate is improved by including error terms for the uncertainty in the profile
shape, and the full temperature and water vapour profiles. The IASI-NN version uses the fixed profiles that were
described by Van Damme et al., (2014) but allows for the use of third party profiles to improve the
representation of the $NH_3$ atmospheric profile. The IASI-LUT and IASI-NN retrievals have both been
previously compared with FTIR observations (Dammers et al., 2016a, Dammers et al., 2016b). They compared
reasonably well with correlations around r=0.8 for a set of FTIR stations, with an underestimation of around
30% that depends slightly on the magnitude of total column amounts,  with the IASI-NN performing slightly
better.

## 2.4     Data criteria & quality

$NH_3$ concentrations show large variations both in space and time as the result of the large heterogeneity in
emission strengths due to spatially variable sources and drivers such as meteorology and land use (Sutton et al.,
2013). This high variability poses challenges in matching ground-based point observations made by FTIR
observations with CrIS downward-looking satellite measurements which have a 14-km nadir footprint. For the
pairing of the measurement data we apply data selection criteria similar to that described in Dammers et al.
(2016a) and summarized in Table 2. To minimize the impact of the heterogeneity of the sources, we choose a
maximum of 50 km between the centre points of the CrIS observations and the FTIR site location. To diminish
the effect of temporal differences between the FTIR and CrIS observations a maximum time difference of 90
minutes is used. Topographical effects are reduced by choosing a maximum altitude difference of 300 m at any
point between the FTIR site location and the centre point of the satellite pixel location. The altitude differences
are calculated using the Space Shuttle Radar Topography Mission Global product at 3 arc-second resolution
(SRTMGL3, Farr et al., 2007). To ensure the data quality of CrIS-$NH_3$ retrieval for Version 1.0, a small number
of outliers with a maximum retrieved concentration above 200 ppb (at any point in the profile) were removed
from the comparison dataset. While potentially a surface $NH_3$ value of 200 ppb (and above) would be possible
(i.e. downwind of forest fires), it is highly unlikely to occur over the entire footprint of the satellite instrument.
Moreover, after inspecting these data points, they seem to be affected by numerical issues in the fitting
procedure (possibly due to interfering species). As we are interested in validating the CrIS observational
information (not just a-priori information), we only select comparisons that contain some information from the
satellite (degrees of freedom for signal (DOFS) ≥ 0.1). Do note that on average the observations have a DOFS
between 0.9 and 1.1. The DOFS > 0.1 filter only removes some of the outliers at the lower end. No explicit filter
is applied to account for clouds; however, clouds will implicitly be accounted for by the quality control as CrIS
will not measure a $NH_3$ signal (e.g. DOFS < 0.1) below optically thick clouds (e.g. cloud optical depth >~1). In
addition, the CrIS observations are matched with FTIR observations taken only during clear-sky conditions,
which mostly eliminates influence from cloud cover. Finally, the high signal to noise ratios (SNR) of the CrIS
instrument, allows it to retrieve $NH_3$ from a thermal contrast approaching 0 K during daytime observations





(Clarisse et al., 2010). Given this, we decided not to apply a thermal contrast filter to the CrIS data. No
additional filters are applied to the FTIR observations beyond the clear-sky requirement.

For both IASI retrievals, we use the same observation selection criteria as described in Dammers et al. (2016a).
The set of criteria is similar to those used here for the CrIS observations. Observations from both IASI retrievals
are matched using the overpass time, and longitudinal and latitudinal positions. For comparability with CrIS a
spatial difference limit of 50 km limit was used, instead of the 25 km spatial limit used in the previous study.
Furthermore we apply the thermal contrast (> 12K, difference between the temperatures at 1.5 km and the
surface) and Earth skin temperature criteria to the IASI observations to match the previous study.

**Table 2. Coincidence criteria and quality flags applied to the satellite and FTIR data. The third through**
**fifth columns show the number of observations remaining after each subsequent data criteria step and**
**the number of possible combinations between the CrIS and FTIR observations. The first set of numbers**
**indicate the number of CrIS observations within a 1°x 1° degree square surrounding the FTIR site.**

| Filter | Data Criteria | Nr. Obs. | | |
|---|---|---|---|---|
| | | FTIR | CrIS | Combinations |
| **CrIS** | | 15661 | 25855 | |
| Temporal sampling difference | Max 90 min | 1576 | 13959 | 112179 |
| Spatial sampling difference | Max 50 km | 1514 | 3134 | 22869 |
| Elevation difference | Max 300 m | 1505 | 1642 | 9713 |
| Quality flag | DOFS ≥ 0.1 | 1433 | 1453 | 8579 |


**2.5     Observational Operator Application**

To account for the vertical sensitivity and the influence of the a-priori profiles of both retrievals we apply the
observational operator (averaging kernel and a-priori of the retrieval) of the FTIR retrieval to the CrIS retrieved
profiles. The CrIS observations are matched to each individual FTIR observation in time and space following
the matching criteria. The FTIR averaging kernels, a-priori profiles, and retrieved profiles are first mapped to
the CrIS pressure levels (fixed pressure grid, layers are made smaller or cut off for observations above elevation
to fit the fixed pressure grid). Following Rodgers and Connor (2003) and Calisesi et al. (2005) this results in the
mapped FTIR averaging kernel, $A_{ftir}^{mapped}$, the mapped FTIR apriori, $x_{ftir}^{mapped,apriori}$, and the mapped FTIR
retrieved profile, $x_{ftir}^{mapped}$. Then we apply the FTIR observational operator to the CrIS observations using **eq.**
**(1)**.
$\widehat{x}_{CrIS} = x_{ftir}^{mapped,apriori} + A_{ftir}^{mapped}(x_{CrIS} - x_{ftir}^{mapped,apriori})$     **(1)**
$\widehat{\Delta x}_{abs} = \widehat{x}_{CrIS} - x_{ftir}^{mapped}$     **(2)**
$\widehat{\Delta x}_{rel} = (\widehat{x}_{CrIS} - x_{ftir}^{mapped})/(0.5\, x_{ftir}^{mapped} + 0.5\, \widehat{x}_{CrIS})$     **(3)**
where $x_{ftir}^{apriori}$ is the FTIR a-priori profile, $x_{ftir}^{mapped}$ is the interpolated FTIR profile, $A_{ftir}^{mapped}$ is the FTIR
averaging kernel, and $\widehat{x}_{CrIS}$ is the smoothed CrIS profile.




The CrIS smoothed profile $\hat{x}_{CrIS}$ calculated from equation (1) provides an estimate of the FTIR retrieval applied
to the CrIS satellite profile. Next we evaluate both total column and profile measurements.
For the first validation step, following Dammers et al. (2016a), who evaluated the IASI-LUT (Van Damme et
al., 2014a) product, we sum the individual profile ($\hat{x}_{CrIS}$ ) to obtain a column total to compare to the FTIR total
columns. This step gives the opportunity to evaluate the CrIS retrieval in a similar manner as was done with the
IASI-LUT retrieval. If multiple FTIR observations match a single CrIS overpass we also average those together
into a single value as well as each matching averaged CrIS observation. Therefore, it is possible to have multiple
FTIR observations, each with multiple CrIS observations all averaged into a single matching representative
observation. For the profile comparison this averaging is not performed to keep as much detail available as
possible. An important point to make is that this approach assumes that the FTIR retrieval gives a better
representation of the truth. While this may be true, the FTIR retrieval will not match the truth completely. For
readability we assume that the FTIR retrieval indeed gives a better representation of the truth, and in the next
sections will describe the case in which we apply the FTIR observational operator to the CrIS values. For the
tenacious reader we included a similar set of results in the appendix, using the CrIS observational operator
instead of the FTIR observational operator, as the assumption of the FTIR being truth is not exactly right.

**3.    Results and discussion**
**3.1        Total column comparison**

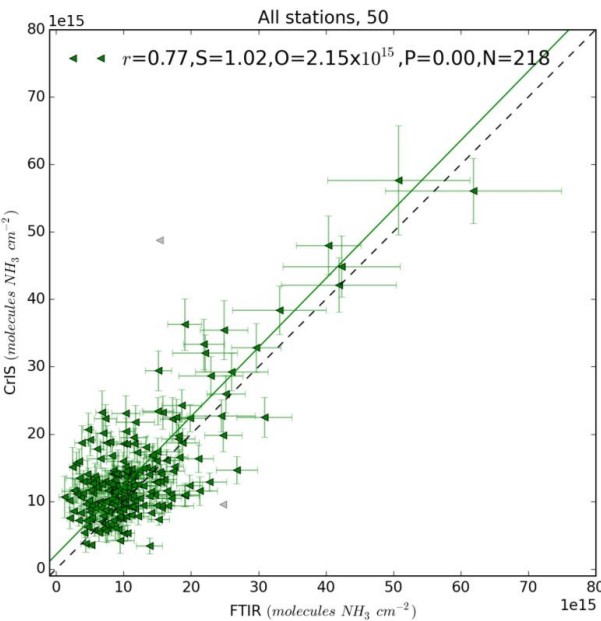


**Figure 2.** Correlation between the FTIR and CrIS total columns using the coincident data from all measurement
sites. The horizontal and vertical bars show the error on each FTIR and CrIS observation. The trend line shows
the results of the regression analysis.



The total columns are averaged as explained in Section 2.4 to show a direct comparison of FTIR measurements
with CrIS observations in Figure 2. A three sigma outlier filter was applied to calculate the regression statistics.
The filtered outliers are displayed in grey, and may be caused by low information content (DOFS) and terrain
characteristics. For the regression we used the reduced major axis regression (Bevington and Robinson, 1992),
accounting for possible errors both in the x and y values. The overall agreement is good with a correlation of r =
0.77 (P < 0.01, N = 218) and a slope of 1.02 (+- 0.05). At the lower range of values the CrIS column totals are
significantly higher than the observed FTIR values. Possibly the CrIS retrieval overestimates due to the low
sensitivity to low concentrations. Without the sensitivity the retrieval will find a value more closely to the a-
priori, which may be too high. Figure 3 shows the comparisons at each station. When the comparisons are
broken down by station (Figure 3), the correlation varies from site to site, from a minimum of 0.28 in Mexico
City (possibly due to retrieval errors associated with the highly irregular terrain) to a maximum of 0.84 in
Bremen. In Toronto, Bremen and Pasadena there is good agreement when $NH_3$ is elevated (> 20 x $10^{15}$
molecules $cm^{-2}$) in, and low bias in the CrIS total columns for intermediate values (between 10 and 20 x $10^{15}$
molecules $cm^{-2}$).

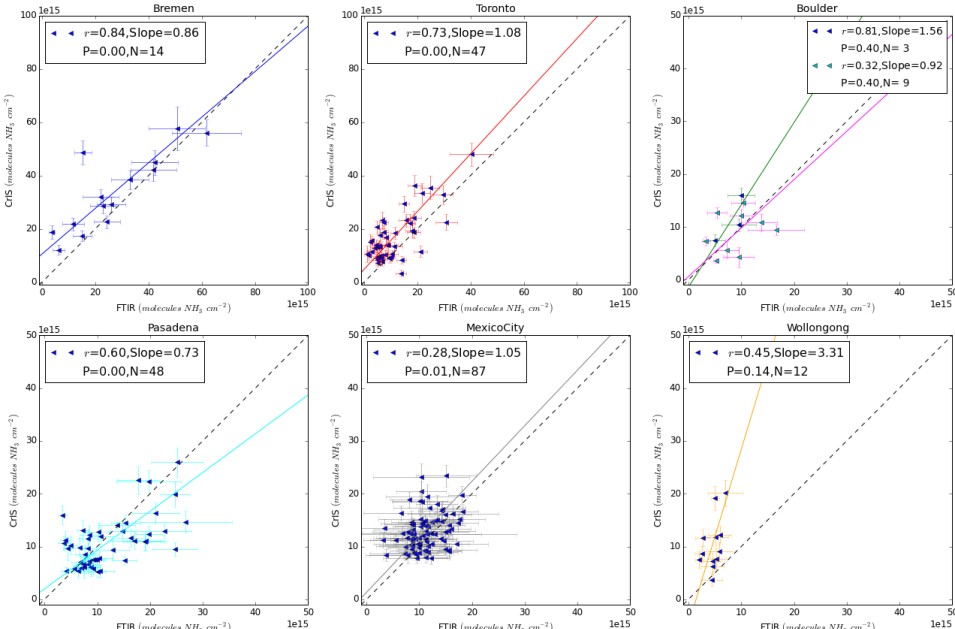


**Figure 3.** FTIR vs CrIS comparison scatter plots showing the correlations for each of the individual stations,
with estimates error plotted for each value. The trend lines show the individual regression results. Note the
different ranges on the x and y axis. The results for the Boulder (green line) and Lauder (pink line) sites are
shown in the same panel.





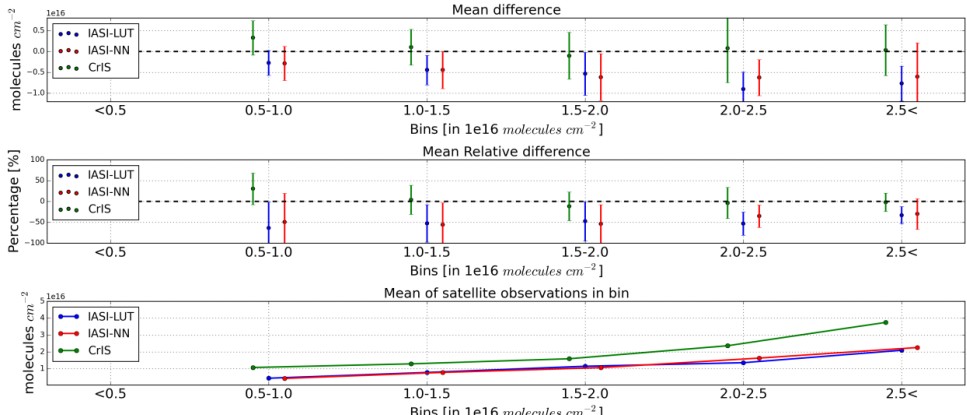

**Figure 4.** Plots of the mean absolute and relative differences between CrIS and IASI, as a function of $NH_3$ total column. Observations are separated into bins of total columns. The upper panel shows the mean absolute difference (MD). The middle panel shows the mean relative difference. The bars in these top two panels show the standard deviation for each value. The bottom panel shows the mean of the observations in each bin.

The mean absolute (MD) and relative difference (MRD) are calculated following equation 4 and equation 5;

$$MRD = \frac{1}{N}\sum_{i=1}^{N}\frac{(CrIS\ column_i - FTIR\ column_i)\ x\ 100}{0.5*FTIR\ column_i + 0.5*CrIS\ column_i} \tag{4}$$

$$MD = \frac{1}{N}\sum_{i=1}^{N}(CrIS\ column_i - FTIR\ column_i) \tag{5}$$

with N being the number of observations.

**Table 3.** Results of the total column comparisons of the FTIR to CrIS, FTIR to IASI-LUT and FTIR to IASI-NN. N is the number of averaged total columns, MD is the mean difference [$10^{15}$ molecules cm$^{-2}$], MRD is the mean relative difference [frac, in %]. Take note that the combined value N does not add up with all the separate sites as observations have been included for FTIR total columns > 5 x $10^{15}$ molecules cm$^{-2}$.

| Retrieval | Column total range in molecules cm$^{-2}$ | N | MD in $10^{15}$ (1σ) | MRD in % (1σ) | FTIR mean in $10^{15}$ (1σ) |
|---|---|---|---|---|---|
| CrIS-NH3 | < 10.0 x $10^{15}$ | 93 | 3.3 (4.1) | 30.2 (38.0) | 7.5 (1.5) |
| CrIS-NH3 | >= 10.0 x $10^{15}$ | 109 | 0.4 (5.3) | -1.39 (34.4) | 16.7 (8.5) |
| IASI-LUT | < 10.0 x $10^{15}$ | 229 | -2.7 (3.0) | -63.6 (62.6) | 7.1 (1.4) |
| IASI-LUT | >= 10.0 x $10^{15}$ | 156 | -5.1 (4.2) | -50.2 (43.6) | 14.8 (6.7) |
| IASI-NN | < 10.0 x $10^{15}$ | 212 | -2.2 (3.6) | -57.0 (68.7) | 7.1 (1.4) |
| IASI-NN | >= 10.0 x $10^{15}$ | 156 | -5.0 (5.1) | -52.5 (49.7) | 14.8 (6.7) |

We evaluate the data by subdividing the comparisons over a set of total column bins as a function of the FTIR total column value of each individual observation. The bins (with a range of 5 x$10^{15}$ to 25 x$10^{15}$ molecules cm$^{-2}$ with iterations steps of 5 x $10^{15}$ molecules cm$^{-2}$) give a better representation of the performance of the retrieval as it shows the influence of the retrieval as a function of magnitude of the total column densities. The results of these total column comparisons are presented in Figure 4. Table 3 summarizes the results for each of the FTIR to satellite column comparisons into two total column bins, which splits the comparisons between smaller and larger than 10 x $10^{15}$ molecules cm$^{-2}$. A few combinations of the IASI-NN and FTIR retrievals have a small





denominator value that causes problems in the calculation of the MRD. A three sigma outlier filter based on the
relative difference is applied to remove these outliers (<10 x $10^{15}$ molecules cm$^{-2}$, only the IASI-NN set). The
statistical values are not given separately by site because of the low number of matching observations for a
number of the sites.

The CrIS/FTIR comparison results show a large positive difference in both the absolute (MD) and relative
(MRD) for the smallest bin, (5.0-10.0 x $10^{15}$ molecules cm$^{-2}$).  The rest of the CrIS/FTIR comparison bins with
NH$_3$ values > 10.0 x $10^{15}$ agree very well with a nearly constant bias (MD) around zero, and a standard
deviation of the order of 5.0 x $10^{15}$ that slightly dips below zero in the middle bin. The standard deviation over
these bins is also more or less constant, and the weak dependence on the number of observations in each bin
indicates that most of the effect is coming from the random error on the observations. The relative difference
becomes systematically smaller with increasing column total amounts, and tend towards zero with a standard
deviation ~25-50%, which is on the order of the reported estimated errors of the FTIR retrieval (Dammers et al,

363     2015).


For a comparison against previous reported satellite results, we included both the IASI-LUT (Van Damme et al.,
2014a) and the IASI-NN (Whitburn et al., 2016) comparisons against the FTIR observations. Both IASI
products show similar differences as a function of NH$_3$ column bins, which is somewhat different from the
CrIS/FTIR comparison results. The absolute difference (MD) is mostly negative with the smallest factor for the
smallest total column bin, with a difference around -2.5 x $10^{15}$ (±3.0 x $10^{15}$) molecules cm$^{-2}$ that slowly increases
as a function of the total column. However, the relative difference (MRD) is at its maximum for the smaller bin
with a difference of the order -50% (±~50%) which decreases to ~ -10-25% (±25%) with increasing bin value.
For both the IASI-NN and IASI-LUT retrievals we find an underestimation of the total columns, which
originates mostly from a large systematic error in combination with more randomly distributed error sources
such as the instrument noise and interfering species, which is similar to results reported earlier for IASI-LUT
(Dammers et al., 2016b).

A number of factors, besides the earlier reported FTIR uncertainties, can explain the differences between the
FTIR and CrIS measurements. The small positive bias found for CrIS points to a small systematic error. The
higher SNR, from both the low radiometric noise and high spectral resolution, along with the shorter
atmospheric path lengths for observations from the ground-based solar-pointing FTIR instrument, enables it to
resolve smaller gradients in the retrieved spectra, which potentially can provide greater vertical information and
detect smaller column amounts (lower detection limit). This could explain the larger MRD and MD CrIS
differences at the lower end of the total column range. However, a number of standalone tests with the FTIR
retrieval showed only a minor increase in the total column following a decrease in spectral resolution, which
indicates that the spectral resolution itself is not enough to explain the difference.

**3.2     Profile Comparison**
The CrIS satellite and FTIR retrieved profiles are matched using the criteria specified above in Table 2 and
compared. It is possible for a CrIS observation to be included multiple times in the comparison as there can be





more than one FTIR observation per day, and /or, the possibility of multiple satellite overpasses that match a
single FTIR observation.
**A representative profile example**
An example of the profile information contained in a representative CrIS and FTIR profile is shown in Fig. 5.
Although the vertical sensitivity and distribution of $NH_3$ differs per station this is a fairly representative. The
FTIR usually has a somewhat larger DOFS in the order of 1.0-2.0, mostly depending on the concentration of
$NH_3$, compared to the CrIS total of ~1 DOFS. Figure 5a shows an unsmoothed FTIR averaging kernel [vmr vmr$^{-1}$
] of a typical FTIR observation. The averaging kernel (AVK) peaks between the surface and ~850 hPa, which is
typical for most observations. In specific cases with plumes overpassing the site, the averaging kernel peak is at
a higher altitude matching the location of the $NH_3$ plume. The CrIS averaging kernel (Fig. 5b) usually has a
maximum somewhere in between 680-850 hPa depending on the local conditions. This particular observation
has a maximum near the surface, an indication of a day with good thermal contrast. Both the FTIR and CrIS
concentration profiles have a maximum at the surface with a continuous decrease that mostly matches the a-
priori profile in shape following the low DOFS. This is visible for layers at the lower pressures (higher altitudes)
where the FTIR and CrIS a-priori and retrieved volume mixing ratios become similar and near zero. The
absolute difference between the FTIR and CrIS profiles can be calculated by applying the FTIR observational
operator to the CrIS profile, as we described in section 2.5. The largest absolute difference (Fig. 5d) is found at
the surface, which is also generally where the largest absolute $NH_3$ values occur. The FTIR smoothed relative
difference (red, striped line) peaks at the pressure where the sensitivity of the CrIS retrieval is highest (~55%),
which goes down to ~20-30% for the higher altitude and surface pressure layers. Overall the retrievals agree
well with most of the difference explained by the errors of the individual retrievals. For an illustration of the
systematic and random errors on the FTIR and CrIS profiles shown in Fig 5, see the figures in the appendix: for
the FTIR error profile see Fig. A1 (absolute error) and A2 (relative error) and for the CrIS measurement error
profile see Fig. A3. Please note that we only show the diagonal error covariance values for each of the errors,
which is common practice. The total column of our example profile is ~20 x 10$^{15}$ molecules cm$^{-2}$ which is a
slightly larger value than average. The total random error is < 10% for each of the layers, mostly dominated by
the measurement error, which is somewhat smaller than average (Dammers et al., 2015) following the larger
$NH_3$ VMR. A similar value is found for the CrIS measurement error with most layers showing an error < 10%.
The FTIR systematic error is around ~10% near the surface and grows to a larger 40% for the layers between
900 – 750 hPa. The error is mostly due to the errors in the $NH_3$ spectroscopy (Dammers et al., 2015). The shape
of the relative difference between the FTIR and CrIS closely follows the shape systematic error on the FTIR
profile pointing to that error as the main cause of difference.





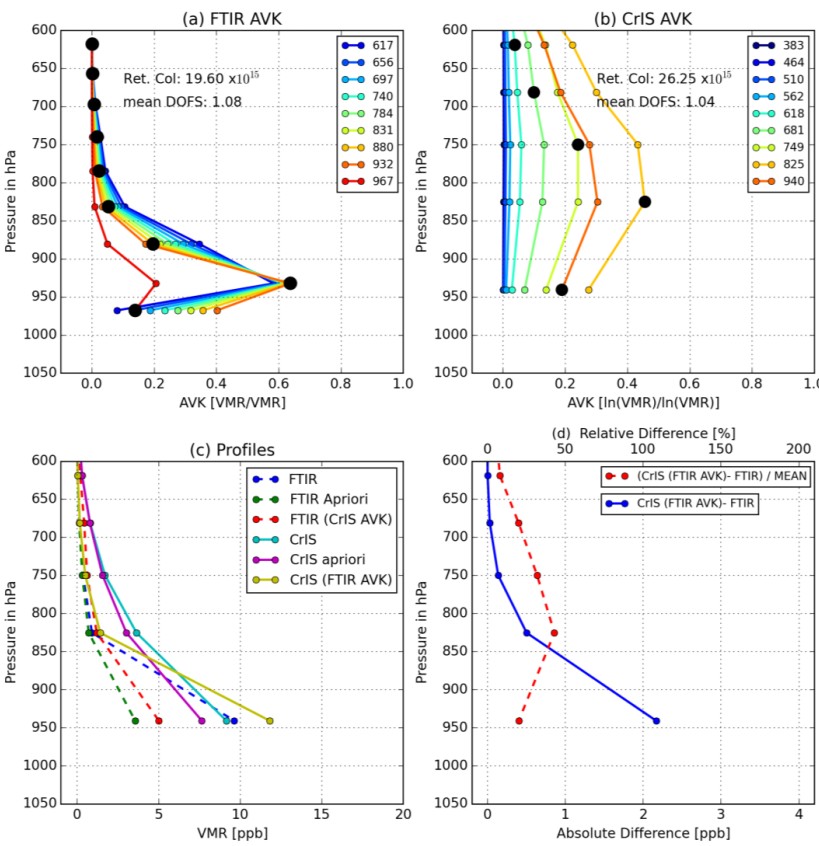

**Figure 5.** Example of the NH$_3$ profile comparison for an FTIR profile matched with a CrIS profile measured around the Pasadena site. With (a) the FTIR averaging kernel, (b) the CrIS averaging kernel. For both averaging kernels the black dots show the matrices diagonal values. Panel (c) shows the retrieved profiles of both FTIR (blue) and CrIS (cyan) with the FTIR values mapped to the CrIS pressure layers. Also shown are the FTIR a-priori (green), the CrIS a-priori (purple), the CrIS retrieved profile smoothed with the FTIR averaging kernel [CrIS (FTIR AVK)] (yellow) and the FTIR profile smoothed with the CrIS averaging kernel [FTIR (CrIS AVK)](red). In panel (d), the blue line is the absolute difference between the FTIR profile (blue, panel (c)) and the CrIS profile smoothed with the FTIR averaging kernel (Yellow, panel (c)) with the red line the corresponding relative difference.

**All paired data**

In Fig. 6 all the individual site comparisons were merged. The Mexico City site was left out of this figure because of the large number of observations in combination with a difference in pressure grid due to the high altitude of the city which obscured the overall analysis and biased the results towards the results of one station. Similar to the single profile example the FTIR profile peaks near the surface for most observations, slowly going towards zero with decreasing pressure. Compared to the representative profile example a number of



differences emerge. A number of FTIR observations peak further above the surface and are shown as outliers
which drag the mean further away from the median values. The combined CrIS profile in Fig. 6 shows a similar
behaviour, although for the lowest pressure layer it has a lower median and mean compared to the layer above.
The difference between Fig. 5 and Fig. 6e derives mostly from the number of observations used in the boxplot,
many with weak sensitivity at the surface. Similar to the single profile example in Fig. 5, the FTIR averaging
kernels in Fig. 6c on average peak near or just above the surface (with the diagonal elements of the AVK's
shown in the figure). The sensitivity varies a great deal between the observations as shown by the large spread
of the individual layers. The CrIS averaging kernels (Fig. 6g) usually peak in the boundary layer around the 779
hPa layer with the 2 surrounding layers having somewhat similar values. The instrument is less sensitive to the
surface layer as is demonstrated by the large decrease in the AVK near the surface, but this varies depending on
the local conditions. We find the largest absolute differences in the lower three layers, as was seen in the
example in Fig. 5, although the differences decrease downwards rather than increase. The relative difference
shows a similar shape to Fig 5. Overall both retrievals agree quite well. The relative differences in the single
level retrieved profile values in Fig. 6h show an average difference in the range of ~20 to 40% with the 25[th] and
75[th] percentiles at around 60-80%, which partially follows from our large range of concentrations. The absolute
difference shows an average difference in the range of -0.66 to 0.87 ppb around the peak sensitivity levels of the
CrIS observations (681 to 849 hPa). The lower number of surface observations follow from the fact that only the
Bremen site is located at an altitude low enough for the CrIS retrieval to provide a result at this pressure level..
Because of this difference in retrieval layering, the remaining 227 observations mostly follow from matching
observations in Bremen, which is located in a region of significant $NH_3$ emissions.




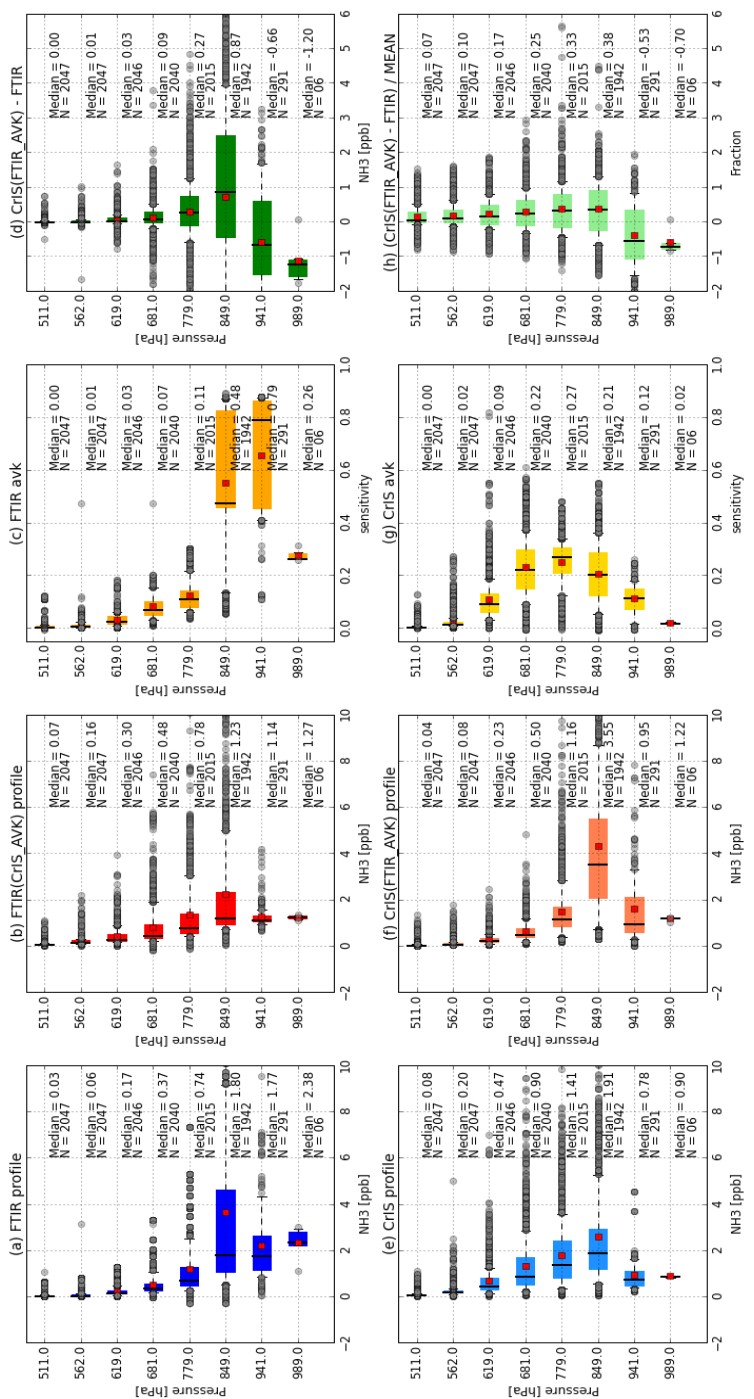


**Figure 6.** Profile comparison for all stations combined. Observations are combined following pressure "bins", i.e. the midpoints of the CrIS pressure grid.
Subplot (a) shows the mean profiles of the FTIR (blue), (b) the profiles of FTIR with the CrIS averaging kernel applied to it (red), (c) the FTIR averaging
kernel diagonal values, and (d) shows the absolute difference [VMR] between profiles (f) and (a). The second row shows the CrIS mean profile in (e), (f) the
profiles of CrIS with the FTIR averaging kernel applied, (g) the CrIS averaging kernel diagonal values, (h) the relative difference [Fraction] between the
profiles in (f) and (a). Each of the boxes edges are the 25th and 75th percentiles, the black lines in each box is the median, the red diamond is the mean, the
whiskers are the 10th and 90th percentiles, and the grey circles are the outlier values outside the whiskers.



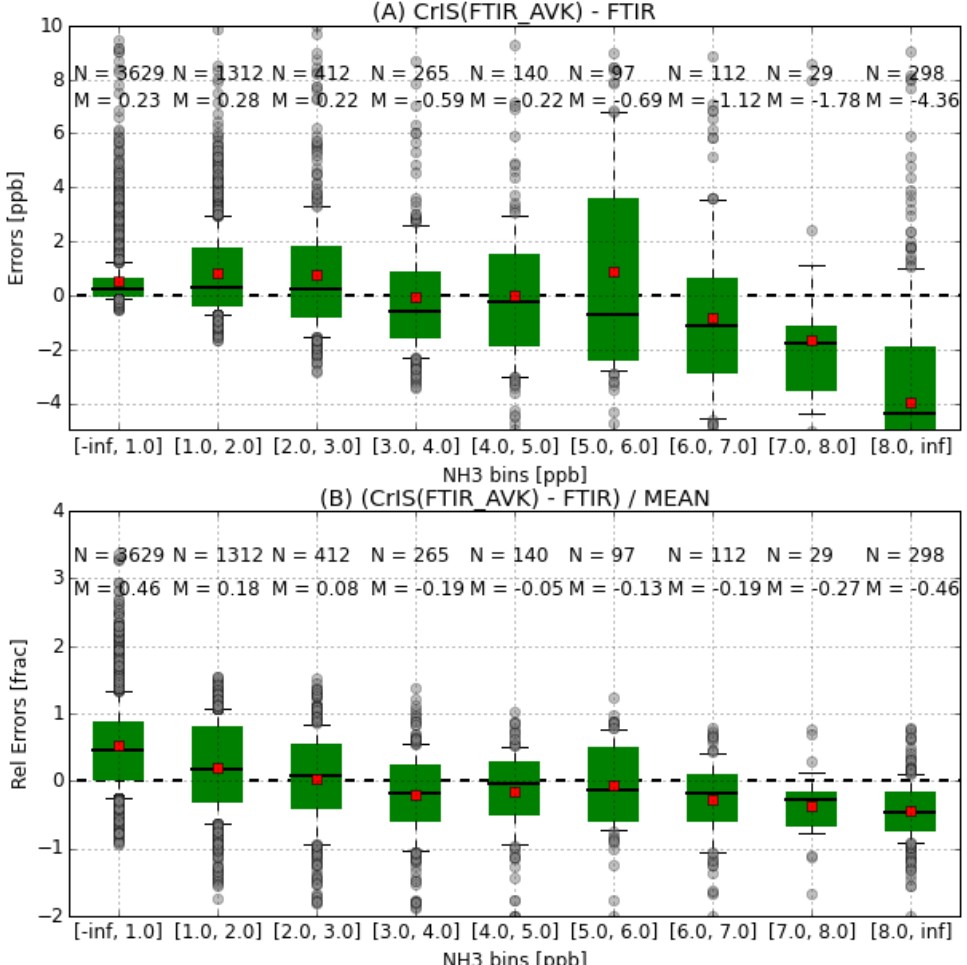

**Figure 7**. Summary of the errors as a function of the VMR of $NH_3$ in the individual FTIR layers. The box edges are the $25^{th}$ and $75^{th}$ percentiles, the black line in the box is the median, the red diamond is the mean, the whiskers are the $10^{th}$ and $90^{th}$ percentiles, and the grey circles are the outlier values outside the whiskers. Only observations with a pressure greater than 650 hPa are used. The top panel shows the absolute difference for each VMR bin, the bottom panel shows the relative difference for each VMR bin.

The switch between negative and positive values in the absolute difference (see Fig. 6d), occurs in the two lowest layers dominated by the Bremen observations and provides insight into the relation between absolute differences as function of retrieved concentration. Fig. 7 shows a summary of the differences as a function of the individual $NH_3$ VMR layer amounts. As seen before in the column comparison, e.g. Fig 2 and 4, the CrIS retrieval gives larger total columns than the FTIR retrieval for the small values of VMR. For increasing VMRs, this slowly tends to a negative absolute difference with a relative difference in the range of 20-30%. However, note that the number of compared values in these high VMR bins are by far lower than in the first three bins





leading to relatively less effect in the total column and merged VMR figures (Figs. 2 and 6) from these high
VMR bins. We now combine the results of Figs. 6 and 7 into Figure 8 to create a set of subplots showing the
difference between both retrieved profiles as a function of the maximum VMR of each retrieved FTIR profile.
For the layers with pressure less than 681 hPa we generally find good agreement, which is expected but not very
meaningful, since there is not much $NH_3$ (and thus sensitivity) in these layers and any differences are smoothed
out by the application of the observational operator. The relative differences for these layers all lie around ~0-
20%. For the lowest two VMR bins we find again that CrIS gives larger results than the FTIR, around the CrIS
sensitivity peak in the layer centred around 849 hPa, and to a lesser extent in the layer below. At these VMR
levels (< 2 ppb) the $NH_3$ signal approaches the spectral noise of the CrIS measurement, making the retrievals
more uncertain. The switch lies around 2-3 ppb where the difference in the SNR between the instruments
becomes less of an issue. Also easily observed is the relation between the concentration and the absolute and
relative differences. This can be explained by the difference in sensitivity of the instruments, and the
measurement noise of both instruments. For the largest VMR bin [> 4.0 ppb] we find that CrIS is biased for the
four lowest layers. Differences are largest in the surface layer where only a few observations are available,
almost all from the Bremen site. Most of these CrIS observations have a peak satellite sensitivity at a higher
altitude than the FTIR. Assuming that most of the $NH_3$ can be found directly near the surface, with the
concentration dropping off with a sharp gradient as a function of altitude, it is likely that these concentrations
are not directly observed by the satellite but are observed by the FTIR instruments. This difference in sensitivity
should be at least partially removed by the application of the observational operator but not completely, due to
the intrinsic differences between both retrievals. The CrIS retrieval uses one of three available a-priori profiles,
which is chosen following a selection based on the strength of $NH_3$ signature in the spectra. The three a-priori
profiles (unpolluted, moderately polluted and polluted) are different in both shape and concentrations. Out of the
entire set of 2047 combinations used in Fig. 8, only six are of the not polluted a priori category. About 1/3 of the
remaining observations use the polluted a-priori, which has a sharper peak near the surface (see Fig. 5c),
compared to the moderately polluted profile, which is used by 2/3s of the CrIS retrievals shown in this work.
Based on the results as a function of retrieved VMR (as measured with the FTIR so not a perfect restriction), it
is possible that the sharper peak at the surface as well as the low a-priori concentrations are restricting the
retrieval. The dependence of the differences on VMR can also possibly follow from uncertainties in the line
spectroscopy. In the lower troposphere there is a large gradient in pressure and temperature and the impact of
any uncertainty in the line spectroscopy is greatly enhanced. Even for a day with large thermal contrast and $NH_3$
concentrations (e.g. Fig 5.), the difference between both the CrIS and FTIR retrievals was dominated by the line
spectroscopy. This effect is further enhanced by the higher spectral resolution and reduced instrument noise of
the FTIR instrument, which potentially makes it more able to resolve the line shapes.


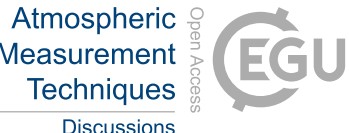

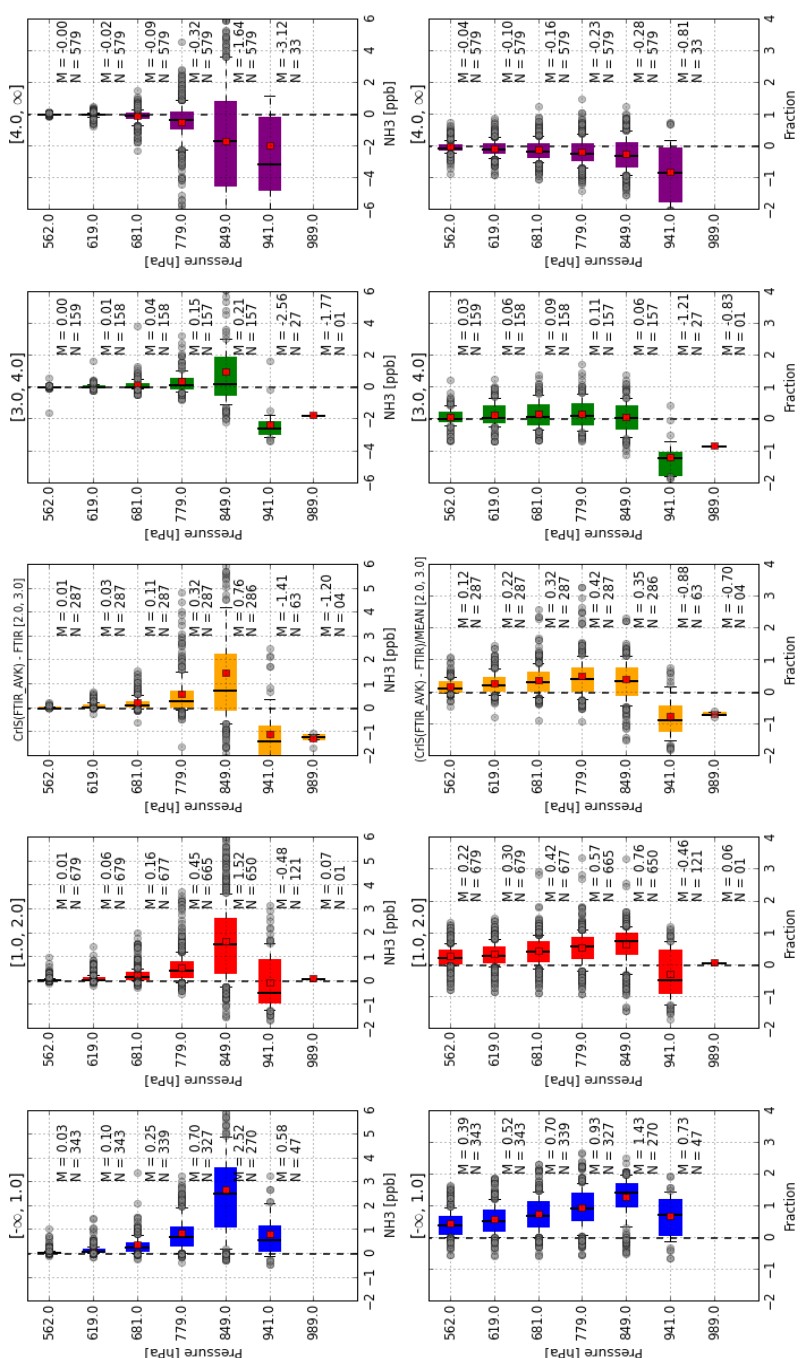

**Figure 8.** Summary of differences as a function of maximum volume mixing ratio (VMR). The maximum VMR of each FTIR profiles is used for the classification. Absolute (Top row) and relative profile differences (bottom row) following the FTIR and CrIS (FTIR AVK applied) profiles. Observations are following pressure layers, i.e. the midpoints of the CrIS pressure grid. The box edges are the 25th and 75th percentiles, the black line in the box is the median, the red diamond is the mean, the whiskers are the 10th and 90th percentiles, and the grey circles are the outlier values outside the whiskers.






To summarise, the overall differences between both retrievals are quite small, except for the lowest layers in the
$NH_3$ profile where CrIS has less sensitivity. The differences mostly follow the errors as estimated by the FTIR
retrieval and further effort should focus on the estimated errors and uncertainties. A way to improve the
validation would be to add a third set of measurements with a better capability to vertically resolve $NH_3$
concentrations from the surface up to ~750 hPa (i.e. the first 2500 m). One way to do this properly is probably
by using airplane observations that could measure a spiral around the FTIR path coinciding with a CrIS
overpass. The addition of the third set of observations would improve our capabilities to validate the satellite
and FTIR retrievals and point out which retrieval specifically is causing the absolute and relative differences at
each of the altitudes.

**4.   Conclusions**

Here we presented the first validation of the CrIS-$NH_3$ product using ground-based FTIR-$NH_3$ observations. The
total column comparison shows that both retrievals agree well with a correlation of R = 0.77 (P < 0.01, N = 218)
and almost no bias with an overall slope of 1.02 (±0.05). For the individual stations we find varying levels of
agreement mostly limited by the small range of $NH_3$ total columns. For FTIR total columns > 10 x $10^{15}$
molecules $cm^{-2}$ the CrIS and FTIR observations agree very well with only a small bias of 0.4 (± 5.3) x $10^{15}$
molecules $cm^{-2}$, and a relative difference 4.57 (± 35.8) %. In the smaller total column range the CrIS retrieval
shows a positive bias with larger relative differences 49.0 (± 62.6) % that mostly seems to follow from
observations near the CrIS detection limit. The results of the comparison between the FTIR and the IASI-NN
and IASI-LUT retrievals, are comparable to those found in earlier studies. Both IASI products showed smaller
total column values compared to the FTIR, with a MRD ~-35- -40%. On average, the CrIS retrieval has one
piece of information, while the FTIR retrieval shows a bit more vertical information with DOFS in the range of
1-2. The $NH_3$ profile comparison shows similar results, with a small mean negative difference between the CrIS
and FTIR profiles for the surface layer and a positive difference for the layers above the surface layer. The
relative and absolute differences in the retrieved profiles can be explained by the errors of the individual
retrievals. Two causes of uncertainty stand out with the $NH_3$ line spectroscopy being the biggest factor, showing
errors of up to 40% in the profile example. The second factor is the signal-to-noise ratio of both instruments
which depends on the VMR: under large $NH_3$ concentrations, the FTIR uncertainty in the signal is in the range
of 10%; for measurements with small $NH_3$ concentrations this greatly increases. Future work should focus on
improvements to the $NH_3$ line spectroscopy to reduce the uncertainty coming from this error source.
Furthermore an increased effort is needed to acquire coincident measurements with the FTIR instruments during
satellite overpasses as a dedicated validation effort will greatly enhance the number of available observations.
Furthermore, a third type of observations measuring the vertical distribution of $NH_3$ could be used to compare
with both the FTIR and CrIS retrievals and further constrain the differences. These observations could be
provided by an airborne instrument flying spirals around an FTIR site during a satellite overpass.



**5.   Data availability**

FTIR-NH$_3$ data (Dammers et al., 2015) can be made available on request (M. Palm, Institut für Umweltphysik,
University of Bremen, Bremen, Germany). The CrIS-FRP-NH$_3$ science grade (non-operational) data products
used in this study can be made available on request (M. W. Shephard, Environment and Climate Change
Canada, Toronto, Ontario, Canada).The IASI-NH$_3$ product is freely available at http://www.pole-
ether.fr/etherTypo/index.php?id=1700&L=1 (Van Damme et al., 2015a).

**6.   Appendix A.**

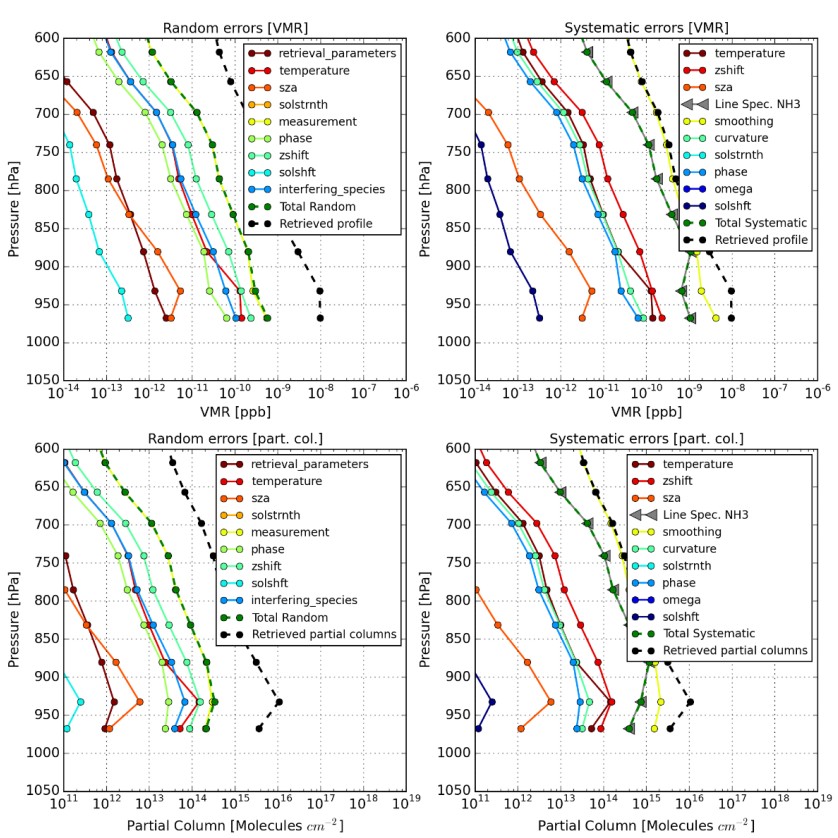

**Figure A1.** Error profiles for each of the error terms. Top panels show the random errors, bottom panels the
systematic errors. Left two panels show the error in VMR. Right panels show the errors in partial column layers
[molecules cm$^{-2}$]. (See Figure A.2 for the same figure but with the errors relative to the final VMR and partial
columns per layer)





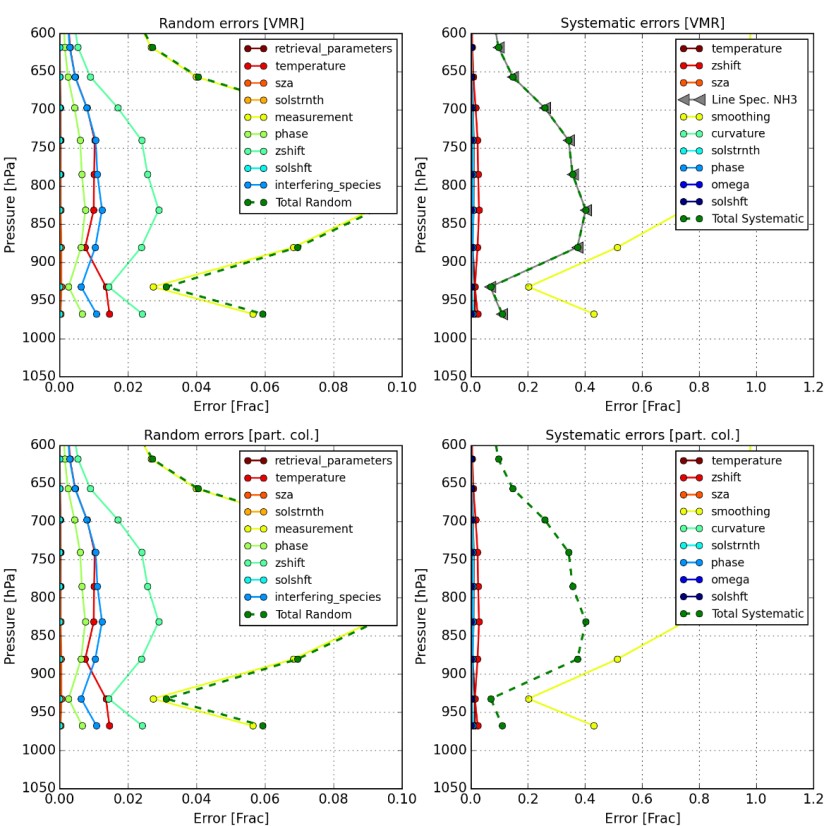


**Figure A2.** Relative error profiles for each of the error terms. Top panels show the Random errors, bottom

panels the Systematic errors. Left two panels show the error in VMR. Right panels show the errors in partial

column layers [molecules cm$^{-2}$]. (See Figure A.1 for the same figure but with the absolute errors)





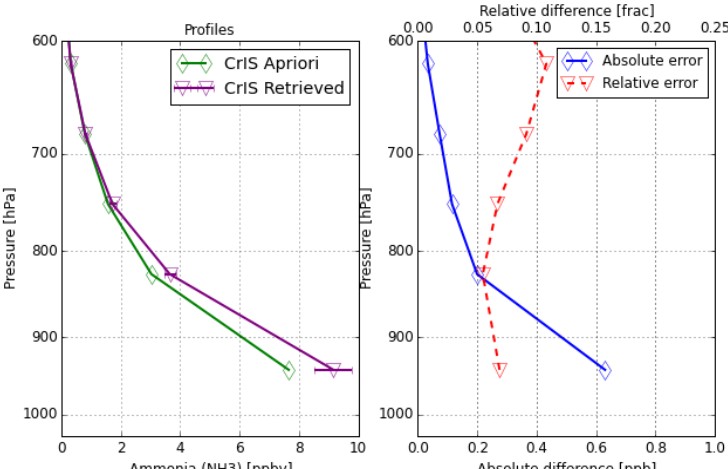


**Figure A3.** CrIS-NH₃ relative and absolute error profile. The left plot shows the retrieved and a-priori profiles

similar to the profiles shown in Figure 5c. The right panel shows the measurement error on the CrIS retrieved

profile, with the blue line the absolute value and red line the value relative to the retrieved profile.


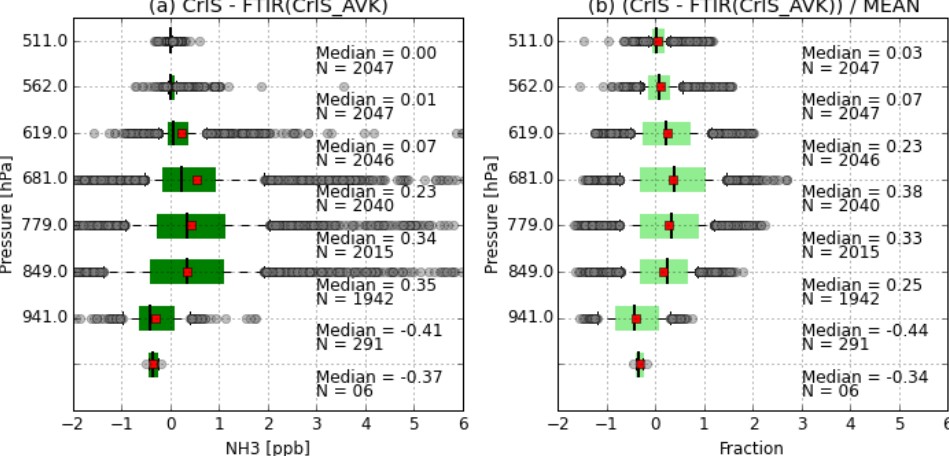

**Fig A4.** Profile comparison for all stations combined. Observations are combined following pressure "bins", i.e.
the midpoints of the CrIS pressure grid. Panel (a) shows the absolute difference [VMR] between profiles (f) and
(a). Panel (b) shows the relative difference [Fraction] between the profiles in (Fig 6e) and (Fig 6b). Each of the
boxes edges are the 25th and 75th percentiles, the black lines in each box is the median, the red diamond is the
mean, the whiskers are the 10th and 90th percentiles, and the grey circles are the outlier values outside the
whiskers.



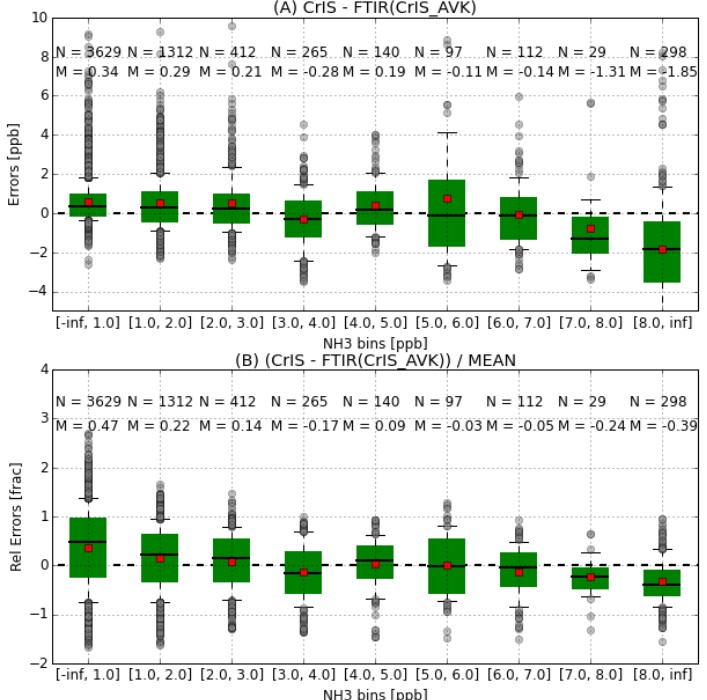

**Fig A5.** Summary of the errors as a function of the VMR of $NH_3$ in the individual FTIR layers. The box edges
are the 25th and 75th percentiles, the black line in the box is the median, the red diamond is the mean, the
whiskers are the 10th and 90th percentiles, and the grey circles are the outlier values outside the whiskers. Only
observations with a pressure greater than 650 hPa are used. The top panel shows the absolute difference for each
VMR bin, the bottom panel shows the relative difference for each VMR bin.





**Fig A6.** Summary of errors as a function of VMR. The maximum VMR of each FTIR profiles is used for the classification. Absolute (Top row) and relative profile differences (bottom row) following the FTIR (CrIS AVK applied) and CrIS profiles. Observations are following pressure layers, i.e. the midpoints of the CrIS pressure grid. The box edges are the 25[th] and 75[th] percentiles, the black line in the box is the median, the red diamond is the mean, the whiskers are the 10[th] and 90[th] percentiles, and the grey circles are the outlier values outside the whiskers.




**Acknowledgements**


This work is part of the research programme GO/12-36, which is financed by the Netherlands Organisation for
Scientific Research (NWO). This work was also funded at AER through a NASA funded
(contract: NNH15CM65C). We would like to acknowledge the University of Wisconsin-Madison Space
Science and Engineering Center Atmosphere SIPS team sponsored under NASA contract NNG15HZ38C for
providing us with the CrIS level 1 and 2 input data, in particular Liam Gumley. We would also like to thank
Andre Wehe (AER) and Jacob Siemons (ECCC) for developing the CrIS download and extraction software. The
IASI-LUT and IASI-NN were obtained from the atmospheric spectroscopy group at ULB (Spectroscopie de
l'Atmosphère, Service de Chimie Quantique et Photophysique, Université Libre de Bruxelles, Brussels,
Belgium) and we would like to thank Simon Whitburn, Martin Van Damme, Lieven Clarisse and Pierre
Francois Coheur for their help and contributions. Part of this work was performed at the Jet Propulsion
Laboratory, California Institute of Technology, under contract with NASA. The University of Toronto FTIR
retrievals were supported by the CAFTON project, funded by the Canadian Space Agency's FAST programme.
Measurements were made at the University of Toronto Atmospheric Observatory (TAO), which has been
supported by CFCAS, ABB Bomem, CFI, CSA, EC, NSERC, ORDCF, PREA, and the University of Toronto.
Funding support in Mexico City was provided by UNAM-DGAPA grants IN107417 & IN112216. A. Bezanilla
and B. Herrera participated in the FTIR measurements and M.A. Robles, W. Gutiérrez and M. García are
thanked for technical support. We would also like to thank Roy Wichink Kruit and Margreet van Marle for the
numerous discussions and valuable input on the subject.

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
