# Peer review of "Validation of the CrIS Fast Physical NH3 Retrieval with"

_Atmospheric Measurement Techniques, 2017_

## Referee Comment (RC1) · Anonymous Referee #1 · 14 Apr 2017

The manuscript presents comparisons between a retrieval product for NH3 of the CrIS satellite instrument ('Fast Physical Retrieval') and ground-based FTIR observations. Both, total column amounts and vertical profiles are intercompared. As knowledge about the global distribution of ammonia is essential for model validation and emission assessments, this work is an important step towards the characterization of the global distributions from space-borne instruments. Unfortunately, there is no comparison with ground-based observations in the region of NH3 'hot-spots' (apart from the Bremen site) like northern India or China which would add important information. In general the applied methods and the results are well described and I support publication of the manuscript. Still there are open issues mainly regarding the total column comparisons as listed below. These should be clarified/implemented before publication.

General comments:

Since the mean number of degrees of freedom of the NH3 retrieval for both instruments seems to be near 1, comparison of the total NH3-column amounts is a central part of the paper. However, some major retrieval characteristics should additionally be provided. Especially a kind of total column operator, like the one shown by Rodgers and Conner, 2003, Fig. 11. E.g. Fig 5 of the actual draft could be modified such that the AK for absolute concentrations or partial column amounts is provided. Further, in Fig. 5 I wonder why the FTIR AK does not peak at the ground level: is there some problem with half-levels there? This should be explained in the paper. Also, while error estimations of the profile retrievals are presented, it would be helpful to have those numbers for the total column amounts as well.

Specific comments:

L27-35:

The abstract should be made more concise. These lines, which include mainly motivation could be skipped.

L39, L48-49: 'compare well'

These are qualitative terms which do not contain much (if any) information content. Please try to avoid those throughout the manuscript and concentrate on quantitative assessments.

LL95-96: 'However, the uncertainty of the satellite observations is still high due to a lack of validation.'

The reasoning is a bit strange: the uncertainty is not caused by lack of validation but rather the knowledge of the uncertainty.

LL245-246: 'Do note that on average the observations have a DOFS between 0.9 and 1.1.'

Could you please provide a Figure or numbers of the DOF distribution of all measurements entering the comparison.

L246: 'clouds will implicitly be accounted for by the quality control':

What is the effect of a partially cloudy field-of-view?

L301: 'total column comparison':

Has this comparison been performed with or without the application of the FTIR-AK as described in chapter 2.5? Since the FTIR is generally better suited for total column retrievals due to its better sensitivity nearer at the ground (where most of the NH3 is present), I doubt that the transformation like in Eq(1), L278 is helpful. Here the better instrument (FTIR) should be transformed to the worse (CrIS) to comparte with the CrIS total column amounts. I.e. there should be Figures like Figs. 2 and 3 with the raw data and after the transformation as just described.

L318: 'In Toronto, Bremen and Pasadena there is good agreement'

In case of Pasadena, I would not call the agreement good. Please also avoid this qualitative terms.

L319: 'and low bias in the CrIS total columns for intermediate values'

This seems not to be the case for Bremen.

L322, Fig. 3:

Could you also discuss in the text what the reason for the apparently systematic deviations at Wollongong may be.

L331, Fig. 4: 'show the standard deviation for each value'

Is this the standard deviation of the distribution of the differences or the standard error of the mean difference (i.e. the former divided by sqrt(number of values))? The latter should (also) be shown to detect any significant measurement bias.

LL379-380: 'along with the shorter atmospheric path lengths for observations from the

ground-based solar-pointing FTIR'

Could you explain, why the FTIR path length is shorter compared to the satellite? Is this always the case?

LL377-385:

As already mentioned, for this discussion the total column operator or the partial column (number density) AK would be interesting. As the FTIR is more sensitive down to the ground level than the satellite where there are highest concentrations of NH3, the satellite retrieval should be determined by the a-priori there. So the higher column amounts may be produced by higher a-priori values at the ground.

L550: 'improvements to the NH3 line spectroscopy to reduce the uncertainty coming from this error source'

Could you give the information if the CrIS retrieval also uses Hitran2012, like the FTIR?

Technical comments:

L566, Fig. A1:

the arrangement of the panels in the figure is transposed with respect to the description in the caption

L571, Fig. A2:

same problem as in Fig. A1. Moreover, the top and the bottom row seem to show identical data.

---

## Referee Comment (RC2) · Anonymous Referee #2 · 19 Apr 2017

The article presents the comparison results between CrIS Fast Physical NH3 retrievals and ground-based remote sensing measurements from FTIR. Given the limited information on the global distribution of CH3, this paper presents an important improvement of our current knowledge. The methodology is sound and after addressing some remaining issues I fully support its publication.

General Comments:

One particular issue that could hamper the interpretation of the results is the potentially limited information content captured by the CrIS retrievals. The current DOFS cut off is taken at >0.1, which entails that some measurements are/could be heavily dominated by the a-priori. The authors allude that particularly measurements with low NH3 concentrations could be effected. One way to at least give some information on this is

to replot Fig2, whereby each measurement is coloured related to its (average) DOFS. Another way to test whether observed differences between CrIS and FTIR are driven by differences in a-priori rather than the actual retrieval, is to (prior to mapping CrIS to FTIR –see eq(1)) conform the CrIS retrieval to the FTIR a-priori as in Rodgers (2000):

x(CrIS,ftir apriori corrected)=x(CrIS) +[A(CrIS)-I]*[apriori(CrIs)-apriori(ftir)]

In any case, the authors need to look deeper into the possible effects of the DOFS on the bias.

A second general comment is the error analysis which could be improved. The document either misses a general statement that all presented errors correspond with the 1-sigma standard deviation or it sometimes needs to be more specific when it uses the term 'error' as it sometimes relates to the standard deviation on the bias and sometimes on the bias itself. That said, 1-sigma standard deviations often tell little with regards to the statistical significance of an observed difference. For instance one claim made by the authors is that in the 0.5-1.0 e16 bin CrIS is significantly higher. This is likely to be true but from the article alone I cannot verify this. In Figure 4 the observed binned biases are shown with their standard deviations. A much better metric to show statistical significance would be the 95 or 99% confidence interval on the mean. This goes for all metrics where statistical significance is claimed or investigated.

Specific comments:

L155: A list of the dominant interfering species would be useful here

L165: A representation of the used collocation area would be useful in this figure

L305: show the error (be more specific= standard deviation is better)

L319: Pasadena looks worse at elevated values

L320: in, and low bias ("in" is obsolete)

L465: red diamond -> red square

L468: Summary of the errors. . . Could be interpreted as the uncertainty on the biases, not the actual absolute and relative bias

---

## Author Comment (AC1) · 21 Jun 2017

We would like to thank Referee #1 for his/her time, constructive and helpful comments and suggestions. Note, throughout the document R1 stands for reviewer 1, and figures named R1.x point to the x'th figure in "Reply to comments, Referee #1.

General comments:

1. Since the mean number of degrees of freedom of the NH3 retrieval for both instruments seems to be near 1, comparison of the total NH3-column amounts is a central part of the paper. However, some major retrieval characteristics should additionally be provided. Especially a kind of total column operator, like the one shown by Rodgers and Conner, 2003, Fig. 11. E.g. Fig 5 of the actual draft could be modified such that

the AK for absolute concentrations or partial column amounts is provided.

We agree it would be a good idea to show total column averaging kernels, but with this version of the CrIS retrieval we did not save the required temperature and water vapour input profiles in the output retrieval files to compute accurate total column averaging kernels. In a future version of the retrieval output files we will compute and provide the total column averaging kernels to go along with the total column values. In the case of the FTIR retrievals, the partial column averaging kernel is provided in the datasets. Figure R1.1 shows the partial column averaging kernel for the example as shown in Figure 5.

2. Further, in Fig. 5 I wonder why the FTIR AK does not peak at the ground level: is there some problem with half-levels there? This should be explained in the paper. The FTIR averaging kernel does not seem to always have complete sensitivity for the ammonia near the surface and varies from observation to observation. As mentioned in the text it usually peaks in between the surface and 850hPa. The method seems to be slightly more sensitive for the second layer in the retrieval. Furthermore, one should take into account that we only have a total DOF of 1 for most observations. Hence one cannot expect a perfect AVK peaking at its own level. We point the referee to the Figures R2.1-R2.4 in our reply to Referee #2.

3. Also, while error estimations of the profile retrievals are presented, it would be helpful to have those numbers for the total column amounts as well. The estimated errors on the FTIR total column amounts are mentioned in section 2.2, being in the order of 30% for which we point you onward to Dammers et al., (2015). In case of CrIS we do not mention a specific percentage in the text, but most total columns have an estimated error in the order of 10 %. This estimate however is on the low side as it does not yet include an estimate for the systematic errors in the retrieval.

Specific comments:

4. L27-35: The abstract should be made more concise. These lines, which include

mainly motivation could be skipped. We removed Line 27-35, and edited the abstract to be more concise. Furthermore we made the following edits: Line 44: Added "(<1.0x1016 molecules cm-2)" Line 45: Removed "and the FTIR total columns are smaller than 1.0x1016 molecules cm-2," Line 46: Removed "are small with CrIS showing" Line 47: Added "show" Line 47: Removed "around +2.4 x 1015 (standard deviation = ±5.5 x 1015) molecules cm-2, which corresponds to a relative difference of ∼+50% (std = ±100 %)." Line 48: Added "The CrIS and FTIR profile comparisons differences are mostly within the range of the estimated retrieval uncertainties single level retrieved profile values showing average difference in the range of ∼20 to 40%" Line 50: Removed "for these comparisons" Line 51: Added "into the boundary layer that typically peaks at" Line 51: Removed "to" Line 52: Added "(∼1.5 km)" Line 52: Removed "and" Line 52: Removed "retrieved profiles also compare well with the" Line 53: Added "is" Line 53: Removed "of" Line 53: Added "std =" Line 53 Added "," Line 53 Removed "and a" Line 53: Added "%" Line 53: Added "std =" Line 54: Removed "Most of the absolute and relative profile comparison differences are in the range of the estimated retrieval uncertainties. However, t" Line 56: Added "At the surface, where CrIS typically has lower sensitivity," Line 55: Removed "he CrIS retrieval does" Line 55: Added "it" Line 56: Added "s" to "tends" Line 56: Removed "the concentrations in the levels near the surface at" Line 56: Added "under" Line 56: Added "conditions, and underestimate under higher atmospheric concentration conditions." Line 58: Removed ", most probably due to the detection limit of the instrument, and at higher concentrations shows more of an underestimation of"

We also made a number of small edits to improve the readability of the main text: Line 25: Edited the email address as the old one is no longer viable (change of institute) Line 71: Added "," Line 97: Added "can" Line 97: Removed "and" Line 107: Added "," Line 110: Added "," Line 192: Changed pseudo-lines to Cross-sections Line 442: Removed "which" Line 443: Added "," Line 445: Added "," Line 462: Removed "." Line 463: Removed "Because of" Line 463: Added "Due to" Line 608: Removed "and Jacob Siemons (ECCC)"

5. L39, L48-49: 'compare well' These are qualitative terms which do not contain much (if any) information content. Please try to avoid those throughout the manuscript and concentrate on quantitative assessments. Removed qualitative terms throughout the document. Line 30 (all line statements are the positions within the new document): changed "compare well with" to "have a positive". Line 301: changed "The overall agreement is good" to "There is an overall agreement" Line 391: changed "good" to "high" Line 400: "removed well" Line 440: changed "agree quite well" to "show agreement" Line 473: removed "good" Line 523: changed 'agree well with" to " have" Line 526: changed "agree very well" to "are in agreement"

6. LL95-96: 'However, the uncertainty of the satellite observations is still high due to a lack of validation.' The reasoning is a bit strange: the uncertainty is not caused by lack of validation but rather the knowledge of the uncertainty.

Edited the sentence to "However, the overall quality of the satellite observations is still highly uncertain due to a lack of validation."

7. LL245-246: 'Do note that on average the observations have a DOFS between 0.9 and 1.1.' Could you please provide a Figure or numbers of the DOF distribution of all measurements entering the comparison.

Figure R1.2 shows the distribution of the DOF of all measurements. Note that the <0.1 DOF are already removed from this set. ∼80 % of the observations have a DOF in between 0.9 and 1.1, with a median of almost 1.0.

8. L246: 'clouds will implicitly be accounted for by the quality control': What is the effect of a partially cloudy field-of-view? That's a good point. Currently, there is a cloud filter in development to exclude clouded scenes in the future. In our case we remove all observations with a DOF of <0.1 which removes most of the clouded scenes (e.g. thick clouds → no ammonia observed). Besides a reduction in DOF we do not expect further major impacts as mentioned in the TES-NH3 retrieval paper (Shephard et al., 2011). As an example to illustrate potential effects (or the lack there of) we give Fig R1.3.

[Figure]

The figure shows a MODIS scene for northern Canada with both visible clouds and fire plumes. The bottom panel shows the calculated CrIS surface NH3 for the same period. As one can observe there are a number of hotspots for NH3 found for the observations surrounding the fires and of the plumes. The optically thick clouds are filtered out by our artificial cut off. The remaining retrieved concentrations for observations with partially covered and by optically thin clouds do not show any strange patterns or alternating high and low retrieved concentrations.

9. L301: 'total column comparison': Has this comparison been performed with or without the application of the FTIR-AK as described in chapter 2.5? Since the FTIR is generally better suited for total column retrievals due to its better sensitivity nearer at the ground (where most of the NH3 is present), I doubt that the transformation like in Eq(1), L278 is helpful. Here the better instrument (FTIR) should be transformed to the worse (CrIS) to compare with the CrIS total column amounts. I.e. there should be Figures like Figs. 2 and 3 with the raw data and after the transformation as just described.

The total column comparison has been performed with the application of the FTIR-AK as described in chapter 2.5. In principle we agree that the better instrument should be transformed to the worse (CrIS). However, we wanted to keep the study comparable to the IASI validation study, and thus apply the AVK in the same manner as done in that study. The IASI product does not produce an averaging kernel and thus we cannot apply the satellite observational operator in both cases. Furthermore, to meet readers who would rather see it the other way around, we added alternative figures transforming the FTIR profiles with the CrIS AVK, which are shown in figures A5, A6 and A7.

10. L318: 'In Toronto, Bremen and Pasadena there is good agreement' In case of Pasadena, I would not call the agreement good. Please also avoid this qualitative terms. Changed the qualitative terms as mentioned in edit number 5.. We also added some lines on the results at Pasadena and Wollongong, see edit number 13 for the full

description. Removed, "Pasadena",

11. L319: 'and low bias in the CrIS total columns for intermediate values' This seems not to be the case for Bremen. The only outlier for Bremen is the value that is marked as an outlier by the three sigma filter as used in Figure 2. Furthermore the number of observations is too small for any good statistics. Added "except for the outlying observation in Bremen, which is marked as an outlier by our three sigma filter used for Figure 2."

12. L322, Fig. 3: Could you also discuss in the text what the reason for the apparently systematic deviations at Wollongong may be. There are a number of reasons why the Wollongong bias might look higher than the others. The first is the date of observation. The two comparisons with the highest CrIS to FTIR ratio were both made during the end of November in 2012 when there were multiple fires occurring in the surrounding region (GFED4.1s). Possibly the CrIS footprint covers the plumes from the fire, which was not observed by the FTIR due to an (for us) unfavourable wind direction. The remaining comparisons on average show a MD of $\sim$ 5 x 1015, which is similar to our station wide result. Another explanation might be the difference in observed air masses which can be larger for coastal sites (e.g. Wollongong, and essentially Toronto). Depending on the wind direction there is either clean air coming in from above the lake/ocean which will mean there is a reduction in FTIR observed NH3 while the satellite potentially observes above land. Vice versa observations from the satellite above the ocean/lake can be far lower than the columns observed by the FTIR with a wind direction coming from an inland direction. This heterogeneity is also visible for sites with larger gradients in orography, such as Pasadena and Mexico City.

Line 311: Added "Similarly to Mexico City the comparison also shows an increase in scatter for Pasadena, where the FTIR site is also located on a hill."

Line 315: Added "In Wollongong, there is less agreement between the instruments. There are two comparisons with large CrIS to FTIR ratios while most of the other

comparisons also show a bias for CrIS. For both cases the bias can be explained by the heterogeneity of the ammonia concentrations in the surrounding regions. The two outlying observations were made during the end of November, 2012, which coincides with wild fires in the surrounding region. Furthermore the Wollongong site is located coastally, which will increase the occurrences where one instrument observes clean air from the ocean while the other observes inland air masses."

13. L331, Fig. 4: 'show the standard deviation for each value' Is this the standard deviation of the distribution of the differences or the standard error of the mean difference (i.e. the former divided by sqrt(number of values))? The latter should (also) be shown to detect any significant measurement bias. Fig 4 showed the standard deviation of the distribution of the differences. As noted in the reply to Referee number 2 we edited the figure to show the 95% confidence interval i.e. ∼2x standard error.

14. LL379-380: 'along with the shorter atmospheric path lengths for observations from the ground-based solar-pointing FTIR' Could you explain, why the FTIR path length is shorter compared to the satellite? Is this always the case? In principle the atmospheric path length should be more or less similar. The path length of both instruments vary per location of the site, time of day and field of view of the satellite, but the difference should be more or less near zero.

Line 379: Removed sentence

15. LL377-385: As already mentioned, for this discussion the total column operator or the partial column(number density) AK would be interesting. As the FTIR is more sensitive down to the ground level than the satellite where there are highest concentrations of NH3, the satellite retrieval should be determined by the a-priori there. So the higher column amounts may be produced by higher a-priori values at the ground. That's a good point but for the fact that the application of the observational operator should reduce the effects of the difference in sensitivity and a-priori choice. Any remaining effect of the a-priori is hard to judge without a repeat of the retrieval. What potentially can be

done to further reduce the influence of the a-priori is switching out the a-priori. For a number of examples of the a-priori switch, we point you to figures R2.1 to R2.4 in our reply to Referee number 2.

16. L550: 'improvements to the NH3 line spectroscopy to reduce the uncertainty coming from this error source' Could you give the information if the CrIS retrieval also uses Hitran2012, like the FTIR? The CrIS retrieval also uses HITRAN 2012.

Added: "and uses the HITRAN database (Rothman et al., 2014) for its spectral lines"

Technical comments: L566, Fig. A1: the arrangement of the panels in the figure is transposed with respect to the description in the caption. Good catch, L566 Fig. A1 caption, changed to: "Error profiles for each of the error terms. The left panels show the random errors, the right panels the systematic errors. The top two panels show the error in VMR. The bottom panels show the errors in partial column layers [molecules cm-2]. (See Figure A.2 for the same figure but with the errors relative to the final VMR and partial columns per layer)"

L571, Fig. A2: same problem as in Fig. A1. Moreover, the top and the bottom row seem to show identical data. It is correct that the top and bottom row are showing identical data as the error is initially derived for the VMR value and subsequently applied to the partial columns.

L571 Fig. A2 caption, changed to: "Relative error profiles for each of the error terms. The left panels show the Random errors, right panels the Systematic errors. All four panels show the error in a fraction of the original unit used in Figure A1. (See Figure A.1 for the same figure but with the absolute errors)"

References. Shephard, M. W., Cady-Pereira, K. E., Luo, M., Henze, D. K., Pinder, R. W., Walker, J. T., Rinsland, C. P., Bash, J. O., Zhu, L., Payne, V. H., and Clarisse, L.: TES ammonia retrieval strategy and global observations of the spatial and seasonal variability of ammonia, Atmos. Chem. Phys., 11, 10743-10763, doi:10.5194/acp-11-

10743-2011, 2011.

van der Werf, G. R., Randerson, J. T., Giglio, L., van Leeuwen, T. T., Chen, Y., Rogers, B. M., Mu, M., van Marle, M. J. E., Morton, D. C., Collatz, G. J., Yokelson, R. J., and Kasibhatla, P. S.: Global fire emissions estimates during 1997–2015, Earth System Science Data Discussions, doi:10.5194/essd-2016-62, in review, 2017.

———————————————

[Figure]

**Fig. 1.** FTIR averaging kernel in [molecules cm-2 / molecules cm-2]. The black dots show the matrices diagonal values.

[Figure]

**Fig. 2.** Distribution plots of the DOF of all CrIS observations used in this study. The left panel shows the fraction of all observations for each specific DOF range. The right panel shows a boxplot of the sam

[Figure]

[Figure]

**Fig. 3.** Top panel shows the MODIS image over Canada on the 10th of August 2013. Bottom panel shows the retrieved CrIS surface concentrations for the same day.

---

## Author Comment (AC2) · 21 Jun 2017

We would like to thank Referee #2 for his/her time, constructive and helpful comments and suggestions. Note, throughout the document R1 stands for reviewer 1, and figures named R1.x point to the x'th figure in "Reply to comments, Referee #1.

General Comments:

1. One particular issue that could hamper the interpretation of the results is the potentially limited information content captured by the CrIS retrievals. The current DOFS cut off is taken at >0.1, which entails that some measurements are/could be heavily dominated by the a-priori. The authors allude that particularly measurements with low NH3 concentrations could be effected. One way to at least give some information on

this is to replot Fig2, whereby each measurement is coloured related to its (average) DOFS.

The artificial DOF cut off was chosen to remove observations without information. We added colouring to the scatterplot indicating the average DOF for each CrIS observation, with the colour bar ranging from 0.1 to more than 1.1. See the reply to the comments of ref #1 for a distribution of the DOF for all observations used in this study. Added "The colouring on the scatter indicates the mean DOF of each the CrIS coincident data." to the caption of Figure 2.

2. Another way to test whether observed differences between CrIS and FTIR are driven by differences in a-priori rather than the actual retrieval, is to (prior to mapping CrIS to FTIR –see eq(1)) conform the CrIS retrieval to the FTIR a-priori as in Rodgers (2000): x(CrIS,ftir apriori corrected)=x(CrIS) +[A(CrIS)-I]\*[apriori(CrIS)-apriori(ftir)].

That is a great question and it is something we have tried before submitting. A posteriori switching out the CrIS a-priori for the FTIR a-priori essentially brings us closer to what we want, validating just the observations without an effect from the a-priori. The problem however is the non-linearity in the retrievals (Kulawik et al., 2008). The resulting retrieved profile is not always near/comparable to the initial a-priori shape and amplitude (which in itself shows that the initial choice of a-priori does not greatly influence the retrieval), which makes an a posteriori switch of the a-priori profile troublesome. Without a repeat of the CrIS retrievals it is not possible to distinguish between the effect of the a priori on the retrieval and the effect of the a posteriori switch. The study by Kulawik et al., (2008) showed that the effect does not have to be major as long as the a priori is representative of the final retrieved profile. This however is not always the case in our retrievals. To illustrate we give a number of examples. Fig R2.1 to R2.4 show the effect for a range of atmospheric ammonia concentrations at a variety of sites, starting with situations with medium to large concentrations for Pasadena (Fig R2.1, which is Figure 5 in the main manuscript), Bremen (Fig R2.2) and Toronto(Fig R2.3). Furthermore we added a figure showing the situation when there is not much AMTD
ammonia i.e. Wollongong (Fig R2.4). In the case of Pasadena the difference is only small with a few percent change in the concentrations of the individual layers. For Bremen the difference is larger near the surface, corresponding to the higher concentration levels. Especially near the surface the concentration change is high following the abrupt difference in the a priori shape, which does not have the sharp peak like the CrIS a priori. In the case of Toronto the effect is in the order of 10%, although concentrations around 750hPa, turn negative. In the Wollongong example the relative difference is large, as the retrieved concentrations are low. The large difference shows that a change of 1-2 ppb in the a priori shape and amplitude cannot be seen as a small enough difference to permit an a posteriori change.  $\tilde{A}\check{C}$

3. In any case, the authors need to look deeper into the possible effects of the DOFS on the bias.

We looked into the effects of the DOF, but decided to not put any further emphasis on it in the main manuscript. Essentially most (>80%) of the observations have a DOF between 0.9 – 1.1. From the remaining 20 % the larger number are above >0.7 leaving a small set of observations ( $\sim$ 10%) with a DOF <0.7. Figure R2.5 shows a scatterplot similar to Figure 2 in the main manuscript but now with only observations with a DOF <0.9. As one can see there is no clear relation visible between the amount of scatter and the DOF. About 20% of the observations (N=45 in total) used for Figure 2 have a DOF < 0.9. Table R1(supplement) shows the mean difference and mean relative difference for the observations with a DOF<0.9. For comparability we added the full set of observations to the table, coloured in red. The observations with small DOFs usually are observations with a relatively low ammonia concentration. This makes that we can only really compare the lower range of total columns to our earlier results as there are only 6 observations with a total column > 10 x 1015 molecules cm-2. The observations smaller than  $> 10 \times 1015$  molecules cm-2 have a MD of 2.6 with a std of 4.1 which is comparable to the complete set with 3.3 (std = 4.1). Similarly the MRD is 33.0 %(std = 54.9%) which is also in the same range as the original set's 30.2% (std = 38.0%)).
A few outlying values are observed in Fig R2.5 which show more of a dependency to location than to DOF, as illustrated in Fig R2.6. All of the larger retrieved total columns (both FTIR and CrIS, >10 x 1015 molecules cm-2) are observations from the Toronto measurement site. As noted in the main manuscript the Toronto results are influenced by the local conditions, which increase the heterogeneity in the region. The site is located within the city, further away from the main sources surrounding the city. Furthermore Toronto is located at the edge of Lake Ontario, which increases the differences as for days with wind originating from the south one can expect clean air observed by the FTIR, where the satellite observes the emitted ammonia of the sources outside the city. Similarly for conditions with wind from the north one can expect

4. A second general comment is the error analysis which could be improved. The document either misses a general statement that all presented errors correspond with the 1-sigma standard deviation or it sometimes needs to be more specific when it uses the term 'error' as it sometimes relates to the standard deviation on the bias and sometimes on the bias itself. That said, 1-sigma standard deviations often tell little with regards to the statistical significance of an observed difference. For instance one claim made by the authors is that in the 0.5-1.0 e16 bin CrIS is significantly higher. This is likely to be true but from the article alone I cannot verify this. In Figure 4 the observed binned biases are shown with their standard deviations. A much better metric to show statistical significance would be the 95 or 99% confidence interval on the mean. This goes for all metrics where statistical significance is claimed or investigated.

Some parts are indeed confusing. We cleaned up the text and added a few sentences to clarify what error we are talking about. We define two types of errors, 1. Estimated errors: FTIR & CrIS retrieval & prior knowledge and 2. the actual errors i.e. the mean and mean relative differences that come out of the comparison. In the case of Figure 4, we edited the figure to include the 95 % confidence level as the number of observations were not included. Throughout the text we left it at standard deviation. Optionally one could calculate it directly from the standard deviation and the number of observations.
Line 296: edited caption, "total estimated error" Line 305: removed "significantly" Line 363, 365: added , N = 229) Line 365, 364: added "std =" in between brackets" Line 403: added "estimated" to "estimated error" Line 460: changed "bias" to "actual error" Line 538: added "estimated" Line 527, 529, 530, 531: added "std =" in between brackets ets Line 582: changed "bias" to "actual error" Caption Fig 4. Added "The number of observations in each set is shown in the bottom panel."

Specific comments:

5. L155: A list of the dominant interfering species would be useful here

L155: Changed "i.e. interfering species" to "(i.e. major interfering species such as H2O, CO2, and O3)"

6. L165: A representation of the used collocation area would be useful in this figure.

Adjusted Figure 1 to include 2 circles with radii of 25 and 50 km. Added a second sentence to the caption of figure 1: "The two circles show the collocation area when for radii of 25 and 50 km."

7. L305: show the error (be more specific= standard deviation is better)

L305: Added "total" to indicate it is not a standard deviation of the observations. The bars indicate the mean total error of the combined observations.

8. L319: Pasadena looks worse at elevated values

L319: Removed Pasadena.

9. L320: in, and low bias ("in" is obsolete)

L320: removed "in"

10. L465: red diamond -> red square

Changed "diamond" to "square in all bar plot figures.

**AMTD**
11. L468: Summary of the errors. . . Could be interpreted as the uncertainty on the biases, not the actual absolute and relative bias.

Caption figure 7: Changed "summary of the errors" to "Summary of the absolute and relative bias". L580: Caption Fig. A5. Similarly changed to "Summary of the absolute and relative bias"

Please also note the supplement to this comment: http://www.atmos-meas-tech-discuss.net/amt-2017-38/amt-2017-38-AC2supplement.pdf

**AMTD**
FTIR:CrIS retrieved profiles: 20130709: Pasadena

**Fig. 1.** Example of the effect of switching out the CrIS a priori for the FTIR a priori to the CrIS Retrieved profile, for an FTIR profile matched with a CrIS profile measured around the Pasadena site. For the

Interactive

comment

FTIR:CrIS retrieved profiles: 20130802: Bremen

**Fig. 2.** Example of the effect of switching out the CrIS a priori for the FTIR a priori to the CrIS Retrieved profile, for an FTIR profile matched with a CrIS profile measured around the Bremen site. For the f

Interactive

comment

FTIR:CrIS retrieved profiles: 20120515: Toronto

**Supplement:**

**Table R1.** Results of the total column comparisons of the FTIR to CrIS for observations with a DOF<0.9. N is the number of averaged total columns, MD is the mean difference [$10^{15}$ molecules cm$^{-2}$], MRD is the mean relative difference [frac, in %]. Value for the complete set used in Table 3 and Fig 4 are given in red. Take note that the combined value N does not add up with all the separate sites as observations have been included for FTIR total columns > 5 x $10^{15}$ molecules cm$^{-2}$.

| Retrieval | Column total range in molecules cm$^{-2}$ | N | MD in $10^{15}$ (1σ) | MRD in % (1σ) | FTIR mean in $10^{15}$ (1σ) |
|---|---|---|---|---|---|
| CrIS-NH3 | < 10.0 x $10^{15}$ | 39 | 2.6 (4.1) | 33.0 (54.9) | 6.1 (2.2) |
| CrIS-NH3 | >= 10.0 x $10^{15}$ | 6 | -4.3 (4.4) | -41.6 (44.3) | 14.2 (4.3) |
| CrIS-NH3 | < 10.0 x $10^{15}$ | 93 | 3.3 (4.1) | 30.2 (38.0) | 7.5 (1.5) |
| CrIS-NH3 | >= 10.0 x $10^{15}$ | 109 | 0.4 (5.3) | -1.39 (34.4) | 16.7 (8.5) |

---

## Author Comment (AC3) · 21 Jun 2017

Author comment

We made a number of small edits to the text to improve the readability of the manuscript. The changes made are the following;

Added a figure to the appendix to improve comparability of the results of the IASI and CrIS retrievals.

Line 366: added "To put the results of this study into perspective of the IASI-LUT and IASI-NN products we added Figure A1 to the Appendix, which shows the total column comparison for both products."

[Figure]

Inserted caption:

"Figure A1. Correlation between the FTIR and the IASI-LUT (left, blue) and IASI-NN (right, red) total columns using the coincident data from all measurement sites. The horizontal and vertical bars show the total estimated error on each FTIR and CrIS observation. A three sigma outlier filter was applied to the IASI-LUT dataset and the same observations were removed from the IASI-NN set. Contrary to the earlier study by Dammers et al., (2016a) no thermal contrast filter was applied to the dataset."

We changed the numbering of the other appendix figures to match the new order.

Shortened and slightly edited the abstract for readability.

Line 44: Added "(<1.0x1016 molecules cm-2)".

Line 45: Removed "and the FTIR total columns are smaller than 1.0x1016 molecules cm-2,".

Line 46: Removed "are small with CrIS showing".

Line 47: Added "show".

Line 47: Removed "around +2.4 x 1015 (standard deviation = $\pm$5.5 x 1015) molecules cm-2, which corresponds to a relative difference of $\sim$+50% (std = $\pm$100 %)."

Line 48: Added "The CrIS and FTIR profile comparisons differences are mostly within the range of the estimated retrieval uncertainties single level retrieved profile values showing average difference in the range of $\sim$20 to 40%"

Line 50: Removed "for these comparisons"

Line 51: Added "into the boundary layer that typically peaks at"

Line 51: Removed "to"

Line 52: Added "($\sim$1.5 km)"

Line 52: Removed "and"

Line 52: Removed "retrieved profiles also compare well with the"

Line 53: Added "is"

Line 53: Removed "of"

Line 53: Added "std ="

Line 53 Added ","

Line 53 Removed "and a"

Line 53: Added "%"

Line 53: Added "std ="

Line 54: Removed "Most of the absolute and relative profile comparison differences are in the range of the estimated retrieval uncertainties. However, t"

Line 56: Added "At the surface, where CrIS typically has lower sensitivity,"

Line 55: Removed "he CrIS retrieval does"

Line 55: Added "it"

Line 56: Added "s" to "tends"

Line 56: Removed "the concentrations in the levels near the surface at"

Line 56: Added "under"

Line 56: Added "conditions, and underestimate under higher atmospheric concentration conditions."

Line 58: Removed ", most probably due to the detection limit of the instrument, and at higher concentrations shows more of an underestimation of"

Further small edits readability in main text

Line 25: Edited the email address as the old one is no longer viable (change of institute)

Line 71: Added ","

Line 97: Added "can"

Line 97: Removed "and"

Line 107: Added ","

Line 110: Added ","

Line 192: Changed pseudo-lines to Cross-sections

Line 442: Removed "which"

Line 443: Added ","

Line 445: Added ","

Line 462: Removed "."

Line 463: Removed "Because of"

Line 463: Added "Due to"

Line 608: Removed "and Jacob Siemons (ECCC)"
* * *